# Inhibition of mTORC1 by ER stress impairs neonatal β-cell expansion and predisposes to diabetes in the *Akita* mouse

Yael Riahi[1], Tal Israeli[1], Roni Yeroslaviz[1], Shoshana Chimenez[1], Dana Avrahami[1,2], Miri Stolovich-Rain[2], Ido Alter[1], Marina Sebag[1], Nava Polin[1], Ernesto Bernal-Mizrachi[3], Yuval Dor[2], Erol Cerasi[1], Gil Leibowitz[1]*

[1]The Endocrine Service, The Hebrew University-Hadassah Medical School, The Hebrew University of Jerusalem, Jerusalem, Israel; [2]Department of Developmental Biology and Cancer Research, The Institute for Medical Research Israel-Canada, The Hebrew University of Jerusalem, Jerusalem, Israel; [3]Department of Internal Medicine, Division of Endocrinology, Metabolism and Diabetes, Miller School of Medicine, University of Miami, Miami, United States

**Abstract** Unresolved ER stress followed by cell death is recognized as the main cause of a multitude of pathologies including neonatal diabetes. A systematic analysis of the mechanisms of β-cell loss and dysfunction in *Akita* mice, in which a mutation in the proinsulin gene causes a severe form of permanent neonatal diabetes, showed no increase in β-cell apoptosis throughout life. Surprisingly, we found that the main mechanism leading to β-cell dysfunction is marked impairment of β-cell growth during the early postnatal life due to transient inhibition of mTORC1, which governs postnatal β-cell growth and differentiation. Importantly, restoration of mTORC1 activity in neonate β-cells was sufficient to rescue postnatal β-cell growth, and to improve diabetes. We propose a scenario for the development of permanent neonatal diabetes, possibly also common forms of diabetes, where early-life events inducing ER stress affect β-cell mass expansion due to mTOR inhibition.

DOI: https://doi.org/10.7554/eLife.38472.001

*For correspondence: gleib@hadassah.org.il

**Competing interests:** The authors declare that no competing interests exist.

## Introduction

β-Cell failure is the fundamental pathophysiological factor of both type 1 (T1D) and type 2 diabetes (T2D) (*Cerasi and Luft, 1967*; *Accili et al., 2010*; *Rhodes, 2005*; *Mathis et al., 2001*). There also exist less frequent, monogenic forms of diabetes resulting from loss-of-function mutations in β-cell function genes. An example is proinsulin mutations which lead to proinsulin misfolding, inducing β-cell ER stress and consequently permanent neonatal diabetes, also called mutant-insulin diabetes of the young (MIDY); its animal counterpart is the *Akita* mouse (*Liu et al., 2010*; *Weiss, 2013*). β-Cells have a highly developed endoplasmic reticulum (ER) to cope with the demand to secrete high amounts of insulin. In diabetes, the proinsulin burden on the ER is increased and proinsulin folding is impaired due to altered β-cell redox state, hence leading to accumulation of misfolded proinsulin and consequently to ER stress. Therefore, proinsulin misfolding/ER stress also plays an important role in the pathophysiology of T1D and T2D (*Eizirik et al., 2008*; *Scheuner and Kaufman, 2008*). Clarifying how ER stress leads to β-cell failure in *Akita* diabetes can have important implications for the common forms of diabetes.

**eLife digest** Insulin is a hormone that is crucial for maintaining normal blood sugar levels and is produced by so called β-cells in the pancreas. If the body stops making insulin, or cells stop responding to it, blood sugar levels rise, leading to diabetes. A form of diabetes known as type 1 diabetes, where the body stops making insulin, usually starts in childhood and can sometimes appear during the first six months of life.

Infants affected by this early onset of diabetes have mutations in one copy of the gene that encodes insulin. They can still produce half of the amount of insulin, which should be sufficient to control blood sugar to a certain extent. Instead, insulin production stops almost completely after a few months. Scientists believe that this is because the mutant insulin has a toxic effect on β-cells.

Mutations in the insulin gene affect the structure of insulin. As a result, insulin builds up in the β-cells, which may eventually cause the cells to die. But the mutant insulin might also cause a problem with a molecule called mTORC1, which helps β-cells to grow.

To investigate this further, Riahi et al. used a mouse model of this form of diabetes to study how stress affects β-cells from birth to adulthood. Mutant β-cells slowed down their rate of cell growth and division early after birth, but did not die more frequently. The results also revealed that β-cells had lower levels of mTORC1, which probably is the main cause of the reduced cell division and growth. When mTORC1 levels were boosted experimentally, the β-cells started to grow and produce more insulin.

Understanding β-cell biology and the link between stress and growth, especially early in life, is a key step in understanding diabetes. In a separate study, Balboa et al. found that human β-cells with insulin mutations also had low mTORC1 and struggled to grow. If boosting mTORC1 could rescue β-cell growth in humans, it could lead to new ways to prevent diabetes.

DOI: https://doi.org/10.7554/eLife.38472.002

β-Cell mass is reduced in diabetes (*Rahier et al., 2008*; *Butler et al., 2003*), albeit with very large variation between subjects, even in T1D (*Campbell-Thompson et al., 2016*). Several mechanisms are implicated, including impaired programming of the endocrine pancreas in utero (*Sandovici et al., 2013*; *Alejandro et al., 2014*), increased β-cell apoptosis (*Butler et al., 2003*; *Jurgens et al., 2011*; *Donath et al., 1999*), reduced β-cell proliferation (*Butler et al., 2007*), and dedifferentiation of mature β-cells (*Talchai et al., 2012*). The quantitative contribution of the different mechanisms to β-cell loss in diabetes is controversial. More important, it is uncertain whether β-cell loss precedes the onset of diabetes or develops during later stages of the disease secondary to hyperglycemia, and thus can rather be viewed as a complication of diabetes. β-Cell mass expands rapidly in the newborn and then adjusts to changes in metabolic demand, probably also in humans (*Bonner-Weir et al., 2016*; *Cigliola et al., 2016*). In mice, islet and β-cell numbers are increased more than 3-fold between 10 days of age and adulthood; this is associated with high β-cell replication, which is drastically decreased during adulthood (*Herbach et al., 2011*; *Teta et al., 2005*; *Saisho et al., 2013*). β-Cell mass expansion is mainly mediated *via* proliferation of mature β-cells (*Dor et al., 2004*). It has been recently suggested that insulin demand drives β-cell proliferation via the unfolded protein response (UPR), which senses insulin production. UPR activation during ER stress correlated with and triggered β-cell proliferation in response to glucose, probably through ATF6 (24). Others showed that reducing the proinsulin load by deleting the insulin gene decreased UPR along with increased β-cell proliferation (*Szabat et al., 2016*), suggesting that ER stress is implicated in the regulation of β-cell proliferation.

Herein, we exploited the *Akita* mouse model of diabetes to study how ER stress affects β-cell mass expansion and differentiation during early life. We found that exposure to ER stress during the neonatal period dramatically reduces β-cell growth and functional maturation. This was associated with transient inhibition of the key signaling complex mTORC1 which governs postnatal β-cell growth and differentiation. Impairment of early β-cell growth and maturation leads to permanent β-cell dysfunction with subsequent development of diabetes; restoration of mTORC1 activity in *Akita* neonates was sufficient to prevent β-cell loss and ameliorate diabetes.

## Results

### β-Cell turnover, differentiation and function in adult *Akita* mice

Metabolic state and islet morphometry were analyzed in 2- to 3-month-old *Akita* mice. Adult *Akita* mice develop severe insulin-deficient diabetes with fed blood glucose ~ 400 mg/dl along with a 90% decrease of pancreatic insulin content (*Figure 1—figure supplement 1a–c*). In adult *Akita* mice, β-cell mass was decreased by 70% compared to age-matched controls (*Figure 1a*). We studied whether decreased β-cell mass is mediated via impaired β-cell proliferation, increased apoptosis or dedifferentiation. The rate of β-cell proliferation measured by Ki67 staining was < 1% and similar in control and *Akita* mice (*Figure 1b*). In agreement with a previous study (*Izumi et al., 2003*), there was a slight increase in the number of TUNEL[+] β-cells in *Akita* mice (*Figure 1b*). Most islets contained no or only a single TUNEL[+] β-cell. We counted 2592 β-cells in wild type and 1754 cells in *Akita* mice and found that the frequency of TUNEL[+] cells was 0.1% in *Akita* mice, whereas no TUNEL[+] cells were observed in control mice; the difference between groups was not statistically significant (p=0.1). Thus, the frequency of apoptotic events based on TUNEL was fairly low in *Akita* β-cells. Apoptotic cells are rapidly cleared by macrophages; therefore, the true rate of apoptosis is very difficult to assess in all models of diabetes. We cannot exclude that cumulative low-grade apoptosis throughout life contributes to β-cell loss in adult animals; however, this finding was somewhat surprising, considering the fact that irreparable proinsulin misfolding generates severe ER stress associated with β-cell loss and insulin deficiency mimicking T1D.

Next, we studied by lineage tracing whether β-cell loss results from β-cell degranulation or transdifferentiation. We generated *RIP-Cre:Rosa26-Yfp* reporter mice on the background of wild-type and *Akita* mice, in order to monitor the fate of β-cells in adult animals. We stained pancreatic sections of β-cell reporter mice for insulin, glucagon and somatostatin and quantified the percentage of genetically labeled β-cells (YFP[+]) expressing insulin or non-β-cell hormones (*Figure 1c*). In *Akita* mice, the number of genetically labeled β-cells that stained negative for insulin (INS) increased by 2.6-fold compared with wild-type mice (*Figure 1d*). Part of the YFP[+]/INS[-] cells expressed glucagon or somatostatin (0.3% of YFP[+]/INS[-] cells; 9 out of 3233 cells) in *Akita* compared to 0.04% in controls; 3 out of 8091 cells). The percentage of β-cells expressing transcription factors required for β-cell maturation and function, including PDX-1 and NKX6.1, was decreased in *Akita* mice (*Figure 1—figure supplement 2*). These findings suggest that some degree of β-cell dedifferentiation/reprogramming does take place in diabetic *Akita* mice; nevertheless, 98.7% of genetically labeled *Akita* β-cells remained insulin positive (*Figure 1d*); therefore, these alterations could not explain the 70% decrease in β-cell mass.

Collectively, decreased β-cell mass in diabetic *Akita* mice is not due to alterations in β-cell proliferation, survival or differentiation in adulthood. We therefore assessed β-cell dynamics during the early postnatal period.

### β-Cell loss in *Akita* mice does not occur in utero

Developmental insults during gestation, such as malnutrition, low-protein diet and increased exposure to glucocorticoids, are known to restrict the number of β-cells formed in the fetal pancreas, which is maintained in adulthood (*Alejandro et al., 2014*; *Dumortier et al., 2011*; *Garofano et al., 1998*). We envisioned that proinsulin misfolding in the embryo after the initiation of insulin biosynthesis at day E11 might lead to ER stress with subsequent impairment of β-cell growth in utero. We analyzed β-cell mass, proliferation and apoptosis in *Akita* and control newborns at P1-2. At this stage, *Akita* mice have normal body and pancreatic weight (*Figure 2a–b*) and are strictly normoglycemic (*Figure 2c*). β-Cell mass in *Akita* newborns was similar to that in control mice (*Figure 2d*). Furthermore, β-cell proliferation was approximately 8-fold higher than in adult animals and was similar in *Akita* and control mice (*Figure 2e*); TUNEL[+] β-cells were found neither in control nor in *Akita* mice (n = 2592 control and 1754 *Akita* β-cells were counted). In *Akita* mice, the percentage of NKX6.1 expressing β-cells was similar to that in control mice, whereas there was a small decrease in the percentage of PDX-1 expressing β-cells (*Figure 2f–g*).

Altogether, these findings indicate that in uteroβ-cell development in *Akita* mice is only minimally impaired, and that β-cell loss must occur after birth.

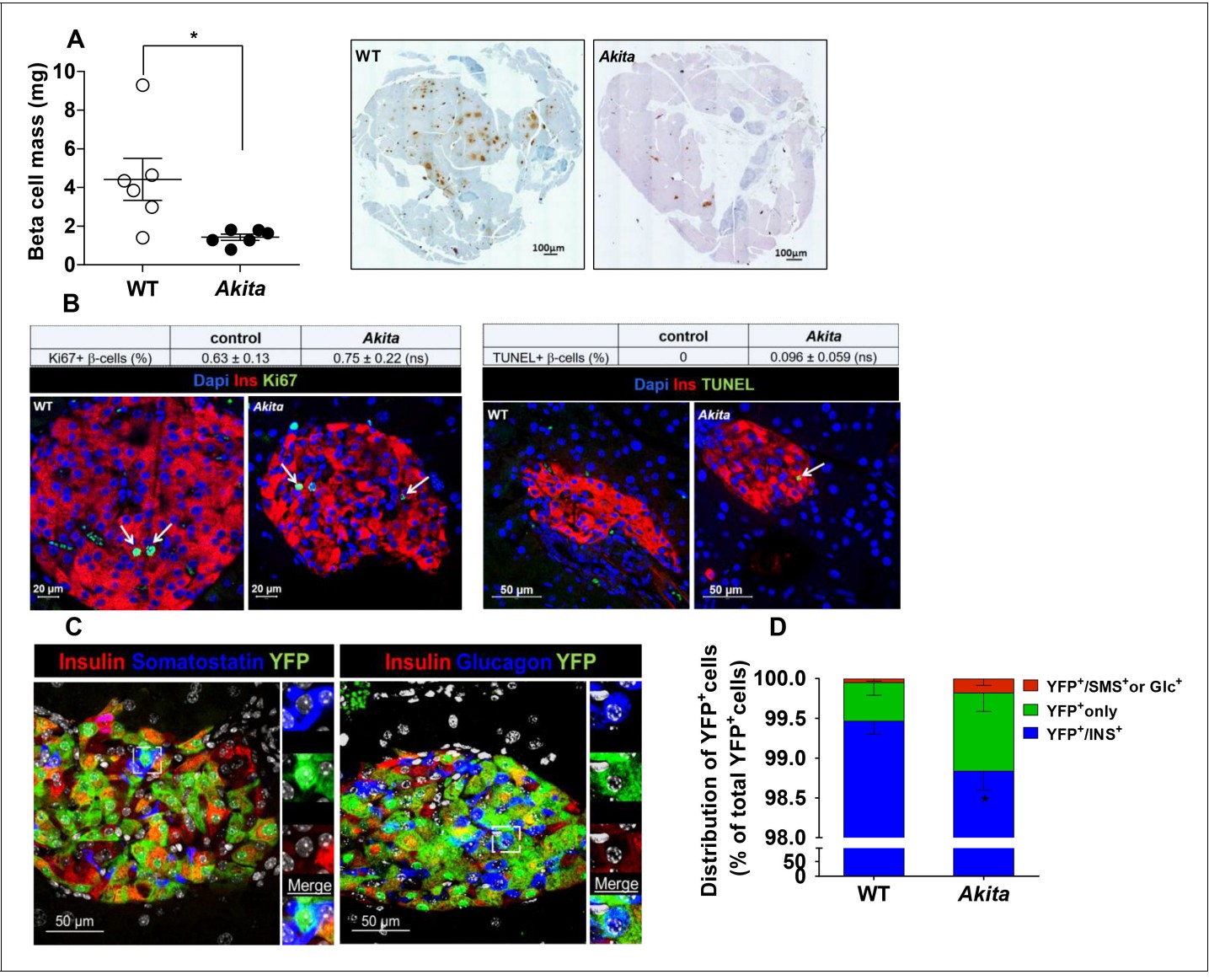

**Figure 1.** β-Cell mass, turnover and differentiation in adult *Akita* mice. Analyses were performed on 2- to 3-month-old *Akita* mice and age-matched controls. (a) β-cell mass (n = 6 in each group); (b) β-cell proliferation and apoptosis assessed by staining for insulin and Ki67 (n = 6–7 mice in each group; a total of 4909 wild type (WT) and 2523 *Akita* β-cells were quantified) or TUNEL (n = 4–5 mice in each group; 2592 WT and 1754 *Akita* β-cells). The percentage of Ki67[+] and TUNEL[+] β-cells is shown in the table above; (c–d) β-cell differentiation was assessed by lineage tracing. Wild-type and *Akita* mice were crossed with *RIP-Cre:Rosa26-Yfp* reporter mice; (c) pancreatic sections of *Akita* mice were immunostained for insulin and somatostatin or glucagon. Lineage-traced β-cells (YFP[+]) expressing somatostatin or glucagon is shown in squares and zoomed in; (d) quantification of insulin-expressing β-cells (percentage of insulin[+]/YFP[+] cells), insulin-degranulated β-cells (percentage of insulin[-]/YFP[+] cells) and of cells with misexpression of somatostatin or glucagon (percentage of somatostatin[+] or glucagon[+]/YFP[+] cells) in WT and *Akita* mice is shown; *p<0.05.

DOI: https://doi.org/10.7554/eLife.38472.003

The following figure supplements are available for figure 1:

**Figure supplement 1.** Glycemia and β-cell function in adult *Akita* and control mice.

DOI: https://doi.org/10.7554/eLife.38472.004

**Figure supplement 2.** NKX6.1 and PDX-1 expression in adult *Akita* β-cells.

DOI: https://doi.org/10.7554/eLife.38472.005

## Impaired β-cell growth during the early postnatal period

We then hypothesized that ER stress might impair β-cell growth during the postnatal period prior to development of diabetes. To test this hypothesis, we assessed the metabolic state and β-cell mass in

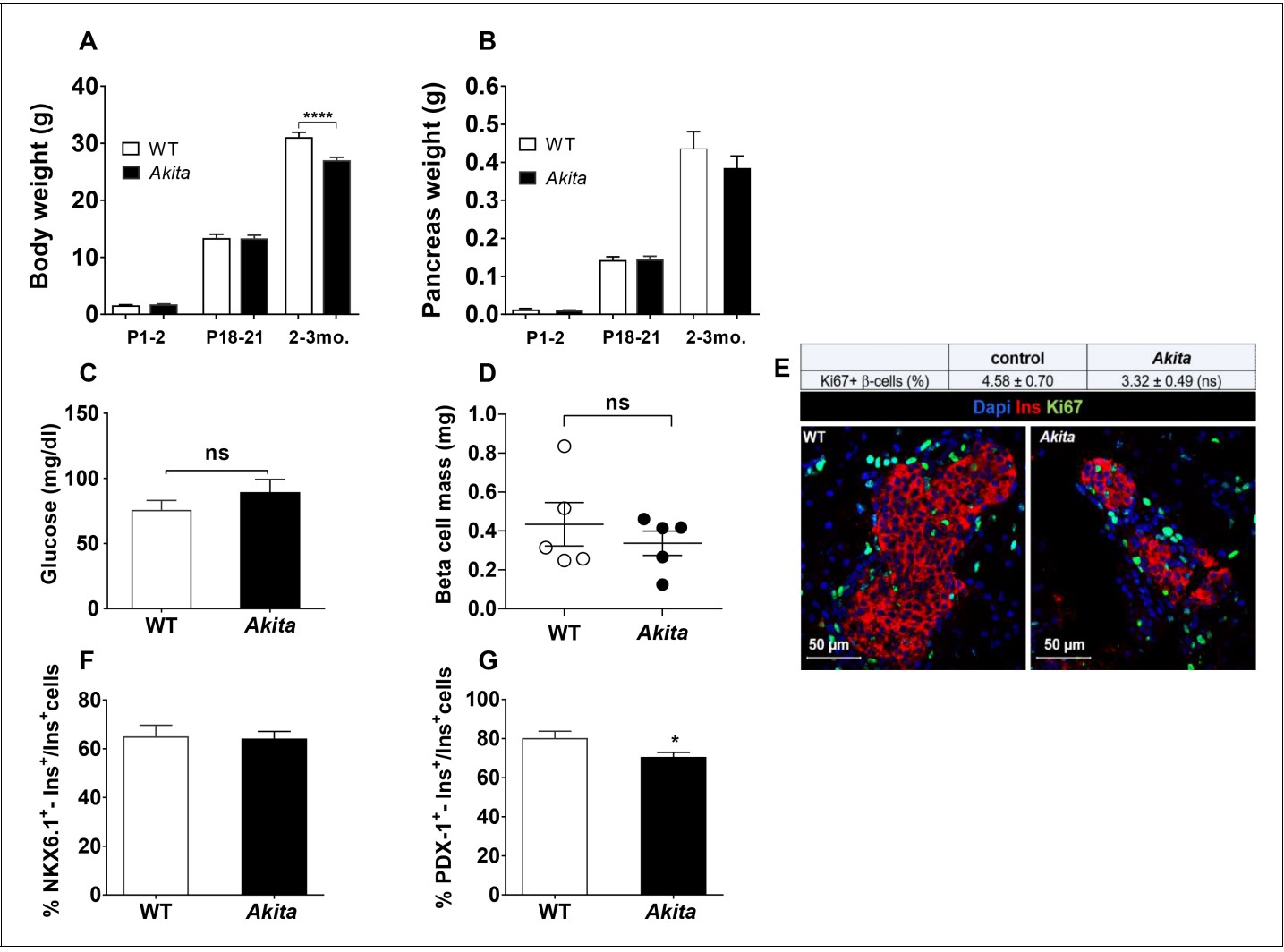

**Figure 2.** Dynamic changes of body and pancreas growth and glycemia, β-cell mass, proliferation and differentiation in *Akita* and control mice at P1-2. (a) body weight, (b) pancreas weight of wild-type and *Akita* mice at P1-2, P19-21 and at the age of 2–3 months. (a) P1-2: WT (n = 8); *Akita* mice (n = 4), P19-21: WT (n = 21); *Akita* mice (n = 23), 2–3 months: WT (n = 33); *Akita* mice (n = 39); (b) P1-2: WT (n = 8); *Akita* mice (n = 4), P19-21: n = 14 in each group, 2–3 months: n = 17 mice in each group). (c) fed blood glucose (n = 7–8 mice in each group); (d) β-cell mass (n = 5 mice in each group); (e) β-cell proliferation assessed by immunostaining for insulin and Ki67 (n = 4 mice in each group; 1886 WT and 1483 *Akita* β-cells). The percentage of Ki67+ β-cells is shown in the table above; (f–g) quantification of β-cells (insulin+) expressing NKX6.1 (n = 3–4 mice in each group; 1148 WT and 1808 *Akita* β-cells) and PDX-1 (n = 3–5 mice in each group; 1364 WT and 1507 *Akita* β-cells). *p<0.05, ****p<0.0001.
DOI: https://doi.org/10.7554/eLife.38472.006

*Akita* compared to control mice at P19-21, prior to weaning. At this stage, body weight and fed and fasting blood glucose in *Akita* mice were still normal (*Figures 2a* and *3a–b*); however, the mice exhibited marked β-cell dysfunction, evident by glucose intolerance associated with blunt insulin response to glucose and decreased pancreatic insulin content (*Figure 3b–d*). Islets isolated from pre-diabetic *Akita* mice also showed marked attenuation of glucose-stimulated insulin secretion along with reduced insulin content (*Figure 3e*). β-Cell mass was decreased by 60% compared to controls, which is similar to the relative decrease in β-cell mass in adult mice (*Figure 3f*). This was accompanied by a parallel decrease in β-cell proliferation based on Ki67, PCNA and phospho-His-tone-H3 immunostaining (*Figure 4a* and *Figure 4—figure supplement 1a*). In control mice, β-cell proliferation remained high in the first 3 weeks of life; this was accompanied by two-fold increase in β-cell mass (*Figure 4d–e*). The decline in β-cell proliferation in *Akita* mice completely prevented the expected early increase of β-cell mass (*Figure 4e*). The proliferation rate in the exocrine tissue was

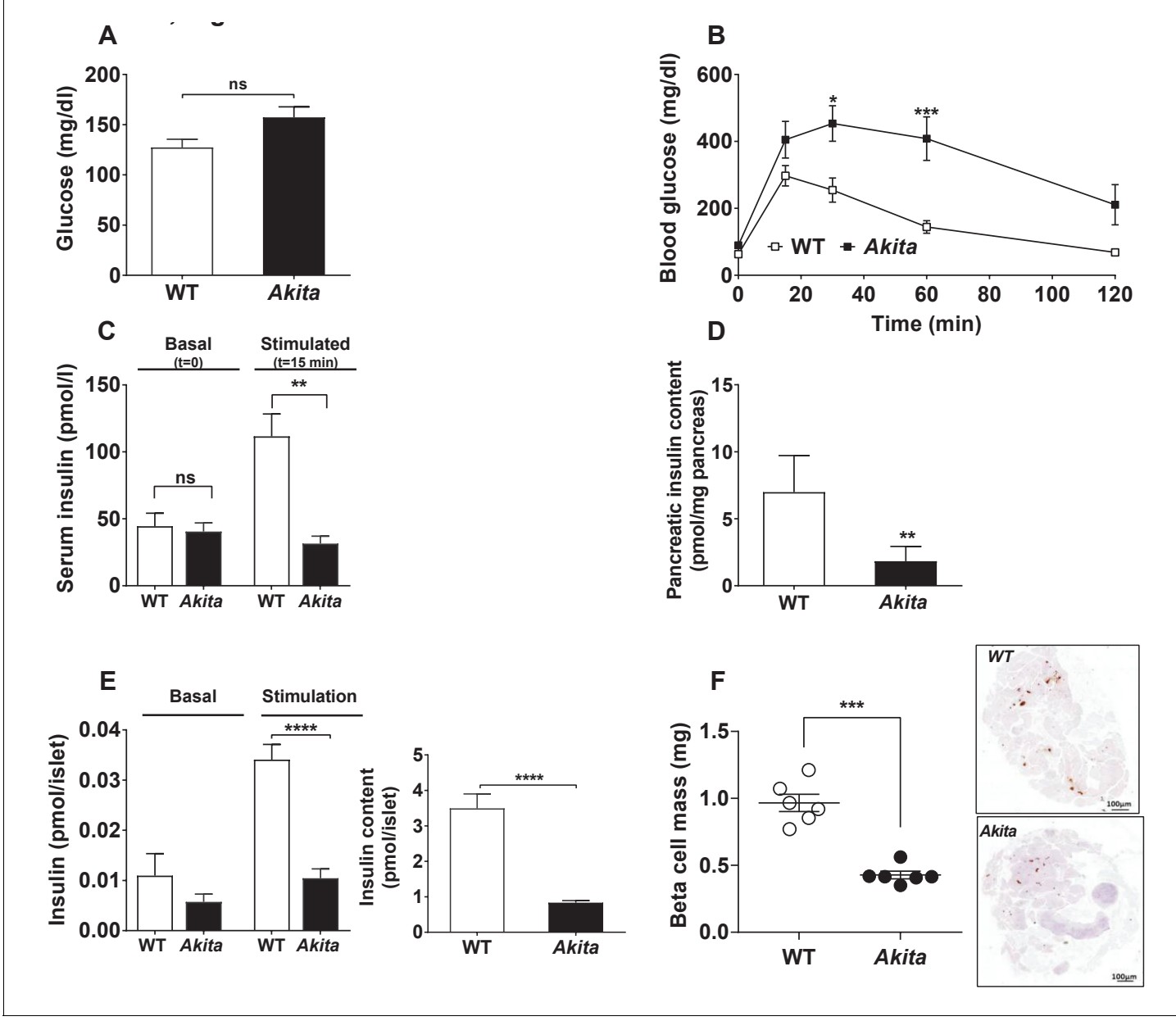

**Figure 3.** Metabolic state, β-cell function and mass in pre-weaning (P19-21) *Akita* mice and age-matched controls. (a) fed blood glucose (n = 7 in each group); (b) IPGTT- glucose (1.5 g/kg) was injected intraperitoneally after an overnight fast (n = 5 in each group); (c) glucose-stimulated insulin secretion in vivo. Insulin was measured before and 15 min following IP glucose injection (1.5 g/kg); (d) pancreatic insulin content (n = 4–5 in each group); (e) basal (3.3 mmol/l glucose) and stimulated (16.7 mmol/l glucose) insulin secretion and insulin content of *Akita* and control islets analyzed by static incubations. Islets were divided into 4 batches of 25 islets per group (n = 3); (f) β-cell mass (n = 6 mice in each group). *p<0.05, **p<0.01, ***p<0.001, ****p<0.0001.
DOI: https://doi.org/10.7554/eLife.38472.007

similar in wild type mice and in *Akita* mice (*Figure 4—figure supplement 1b*), indicating that the effect of the *Akita* mutation on proliferation is cell autonomous. Consistently, the weight of the pancreas, which mainly contains exocrine tissue, was similar in *Akita* and control mice (*Figure 2b*). To assess β-cell size, we used insulin staining to mark β-cells and E-cadherin to highlight cell boundaries (*Figure 4b–c*). In control mice, β-cell size remained unchanged during the first 3 weeks of life and increased 3-fold in adult animals. In *Akita* mice, β-cell size decreased during the early postnatal period, but increased after weaning while developing diabetes. *Akita* β-cells were smaller than control at P1-2, P19-21 and in 2- to 3-month old animals: a 33% reduction in β-cell size was observed in

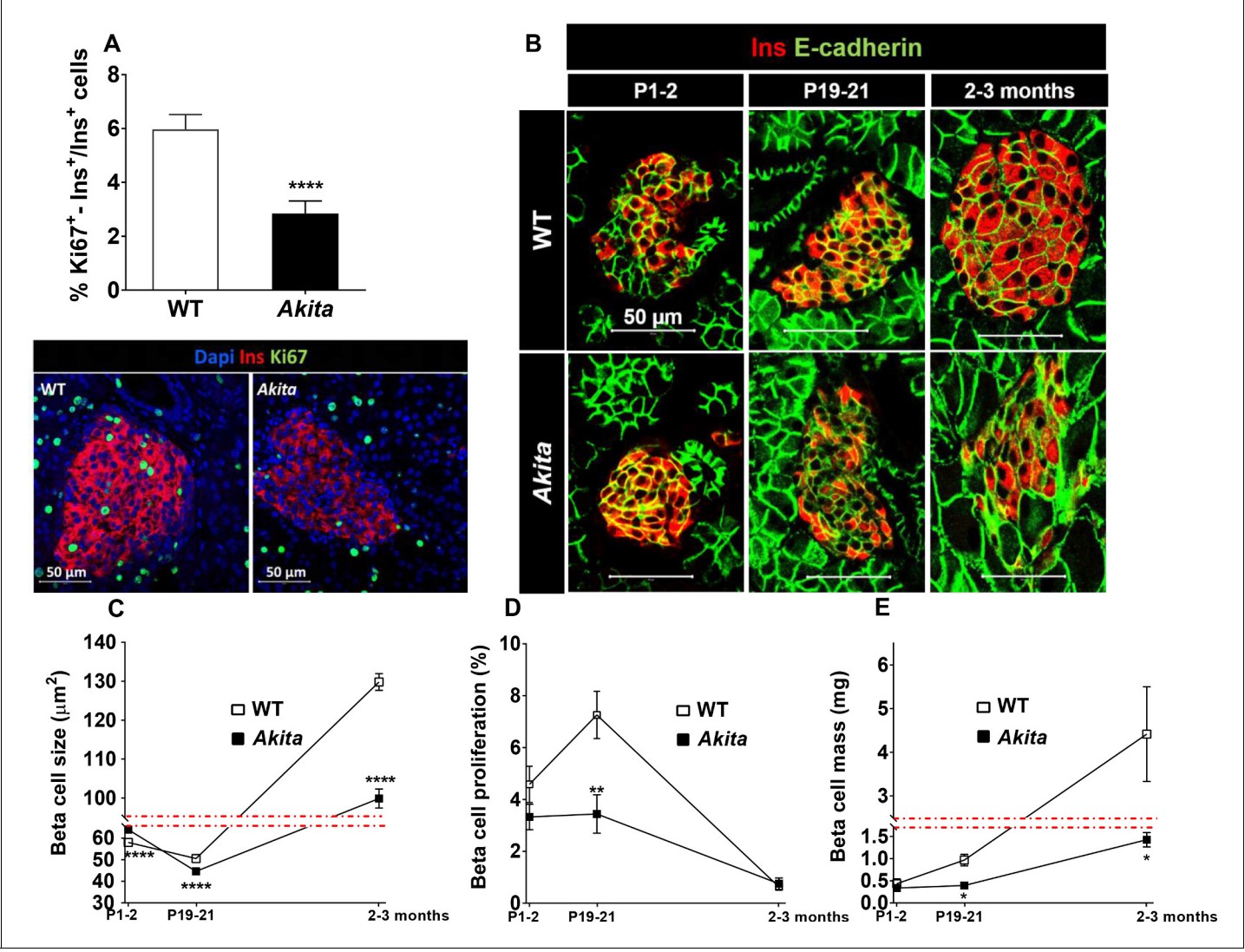

**Figure 4.** Dynamic changes in β-cell expansion in *Akita* and control mice. (a) β-cell proliferation assessed by immunostaining for insulin and Ki67 (n = 6 mice in each group; 2541 WT and 3391 *Akita* β—cells); (b) β-cell size at P1-2 (newborn, n = 4–5 mice in each group; 334 WT and 435 *Akita* β-cells), P19-21 (pre-weaning, n = 3 mice in each group; 330 WT and 364 *Akita* β—cells) and in adult mice (2–3 month-old, n = 3 mice in each group; 266 WT and 417 *Akita* β—cells) assessed by immunostaining for E-cadherin and insulin. Quantifications of β-cell size (c), proliferation (d), and mass (e) are shown. *p<0.05, **p<0.01, ****p<0.0001.

DOI: https://doi.org/10.7554/eLife.38472.008

The following figure supplement is available for figure 4:

**Figure supplement 1.** Proliferation of β-cells and exocrine cells in pre-weaning *Akita* and control mice.

DOI: https://doi.org/10.7554/eLife.38472.009

adult *Akita* mice (**Figure 4b–c**). Notably, β-cell mass increased after weaning both in wild type and in *Akita* mice (**Figure 4e**); however, the overall increase in β-cell mass in *Akita* mice was attenuated compared to controls due to the lower rate of β-cell proliferation and smaller increase in β-cell size. We did not detect any TUNEL[+] β-cells at P19-21, neither in control nor in *Akita* mice (n = 2676 control and 1447 *Akita* β-cells were counted), indicating that apoptotic events were quite rare even in *Akita* mice.

In summary, *Akita* β-cell mass is decreased due to impaired postnatal β-cell growth early in life, prior to the onset of full-blown diabetes.

## Impaired β-cell differentiation and functional maturation in *Akita* neonates

To understand the mechanisms underlying *Akita* β-cell growth arrest and dysfunction prior to development of diabetes, we isolated islets from pre-weaning mice at P19-21, and analyzed gene expression by RNA-seq. It has been previously reported that in heterozygous *Akita* mice β-cell loss is accompanied by decreased α-cell number and that islet composition remained unchanged (*Kayo and Koizumi, 1998*). Consistently, we found that β-cell number per islet area and β/α cell ratio were similar in neonate *Akita* and wild-type mice (*Figure 5—figure supplement 1*), indicating that transcriptomic analysis mainly reflects the changes in the genetic signature of the β-cells (~70% of all islet cells) and is not influenced by alterations in islet composition. The list of differentially expressed genes between *Akita* and control islets is shown in *Table 1*. We performed geneset enrichment and pathway analyses using Genomica and Ingenuity software. ER-stress-related genes were upregulated in *Akita* islets, along with modest enrichment of genes involved in apoptosis (*Figure 5a*). Intriguingly, total steady state mRNA levels of *Xbp1*, the main ER stress-sensing transcription factor, were decreased with only a modest increase in *Xbp1* splicing (*Figure 5b*). The most prominent upregulated UPR gene was Homocysteine-responsive endoplasmic reticulum-resident ubiquitin-like domain member one protein (*Herpud1*) (log 2FC 1.8; p=4.6×10$^{-24}$). HERPUD1 functions as a hub for membrane association of ER associated degradation (ERAD) machinery components and for the interactions between misfolded proteins and ERAD. The expression of chaperones, including *Dnajc3* (*Hsp40*), *Manf* and *Hspa5* (*Bip*) was upregulated in *Akita* islets (*Figure 5b*); the protein level of the latter was also markedly increased (*Figure 6a*). There was a mixed response of genes that regulate apoptosis in ER stress: *Atf6*, *Atf3*, *Ddit3* (*Chop*), *Txnip* and *Bbc3* (*Puma*) were upregulated, whereas *Atf4* and pro-apoptotic *Bax* were downregulated (the changes in *Atf3*, *Atf4*, *Atf6* and *Bax* were not statistically significant) (*Figure 5b*). P85α is a regulatory unit of PI3 kinase; it has been shown that P85α deficiency protects β-cells from ER-stress-induced apoptosis (*Winnay et al., 2014*). In *Akita* islets, the expression of *Pik3r1* gene encoding for P85α was decreased (*Figure 5b*), probably promoting β-cell survival. It has been previously shown that in *Akita* mice ER stress-induced apoptosis is mediated via CHOP; however, CHOP expression was increased only after development of diabetes, but not during the neonatal period (*Oyadomari et al., 2002*), further suggesting that young β-cells adapt to chronic ER stress without robust stimulation of the terminal, pro-apoptotic UPR.

Strikingly, we found that genes regulating β-cell differentiation and function were downregulated in pre-weaning, neonate *Akita* islets (*Figure 5c–d* and *Table 1*). This included the transcription factors *Nkx6.1*, *Nkx2.2* and *Mafa*, proinsulin (*Ins1* and *Ins2*), pancreatic convertase 1/3 (*Pcsk1*) and Glut2 (*Slc2a2*), as well as genes involved in calcium signaling, insulin granule formation and secretion (*Table 1*). Consistently, target genes of NKX6.1 and PDX-1 transcription factors, master regulators of β-cell differentiation and function, were also downregulated, suggesting impairment of β-cell differentiation (*Figure 5c–d*). RNA-seq showed that in *Akita* neonates, the mRNA level of *Pdx1* was not significantly downregulated (*Figure 5b*), whereas PDX-1 protein level was markedly reduced (*Figure 6a*). Immunostaining showed that the number of β-cells expressing NKX6.1 and PDX-1 was decreased by ~50% (*Figure 6b*), indicating that the lower expression of β-cell transcription factors is not due to decreased β-cell number per se. Finally, we treated adult and neonate islets and the β-cell line INS-1E with low-dose thapsigargin; this decreased PDX-1 protein level (*Figure 6c–d*). Chemical chaperones (TUDCA and 4-PBA) had variable effects on BiP expression; however, both compounds failed to prevent the effect of ER stress on PDX-1 expression (*Figure 6c*). Mitochondrial activity has been implicated in cell proliferation including that of β-cells (*Walter et al., 2015*; *Klochendler et al., 2016*) and is instrumental for β-cell functional maturation and for the development of mitogenic and secretory responses to glucose (*Stolovich-Rain et al., 2015*). Several genes encoding subunits of the electron transport chain including *Ndufs2*, *Sdhc*, *Cox6a2*, *Cox6c*, as well as the key anaplerotic enzyme pyruvate carboxylase (*Pcx*) were downregulated in neonate *Akita* islets. On the contrary, the expression of pyruvate dehydrogenase kinases (*Pdk1, 2* and *4*), which inhibit oxidative phosphorylation by phosphorylating pyruvate dehydrogenase, was upregulated (*Table 1*).

In summary, ER stress leads to decreased expression of key β-cell transcription factors and mitochondrial genes along with impaired postnatal β-cell differentiation and functional maturation.

**Table 1.** Transcriptome changes in P19-21 Akita islets compared to age-matched controls (n = 3 per each group).

| Gene symbol | log2 Fold change | p value | Gene symbol | log2 Fold change | p value |
|---|---|---|---|---|---|
| β cell signature | | | Growth factors and mTOR signaling | | |
| Pcsk1 | −2.1101 | 1.1341E-13 | Dapp1 | −2.0554 | 7.7092E-09 |
| Mafa | −1.9801 | 2.1124E-12 | Egfr | −2.0185 | 2.0824E-08 |
| Igsf11 | −1.7632 | 2.8185E-05 | Cth | −1.9445 | 4.9302E-05 |
| Insulin II | −1.7070 | 5.9006E-14 | Igf2 | −1.5846 | 4.3954E-03 |
| Ucn3 | −1.5979 | 2.8193E-06 | Tubg1 | −1.5134 | 3.6446E-06 |
| Nkx6-1 | −1.5542 | 5.0005E-11 | Sqle | −1.4758 | 5.5074E-07 |
| Vdr | −1.4299 | 3.0451E-08 | IGF1r | −1.4735 | 1.5160E-05 |
| Slc2a2 | −1.4352 | 5.3528E-04 | Tpi1 | −1.4516 | 1.4280E-15 |
| Insulin I | −1.3988 | 1.7436E-11 | Btg2 | −1.4350 | 1.5855E-03 |
| Nkx2-2 | −1.2408 | 8.1578E-04 | Elovl6 | −1.3484 | 3.1052E-03 |
| | | | Elovl5 | −1.3484 | 6.7970E-08 |
| Insulin secretion, Insulin granules | | | Mllt11 | −1.3283 | 9.1320E-04 |
| Sytl4 | −2.1903 | 6.6331E-18 | Ppa1 | −1.3162 | 4.9816E-04 |
| Pcsk1 | −2.1101 | 1.1341E-13 | Uchl5 | −1.3085 | 8.1939E-03 |
| Vgf | −2.1003 | 2.1515E-18 | IGF2r | −1.2440 | 1.9131E-05 |
| Gng12 | −1.6254 | 3.0041E-07 | | | |
| Syt5 | −1.5143 | 1.3141E-04 | Cell cycle, replication | | |
| Iqgap1 | −1.4997 | 2.8284E-08 | Pak6 | −2.1602 | 1.0026E-12 |
| Chrm3 | −1.4571 | 1.2136E-05 | Tmem71 | −2.0697 | 3.8133E-12 |
| Gng4 | −1.4365 | 2.7040E-06 | S100a10 | −2.0519 | 9.5186E-07 |
| Chgb | −1.3739 | 1.4523E-03 | Spc25 | −1.8739 | 1.1632E-05 |
| Gpr119 | −1.3712 | 2.6136E-03 | Mpp6 | −1.8457 | 3.7249E-06 |
| Ptprn | −1.3342 | 1.7018E-04 | Plagl1 | −1.8253 | 1.5119E-12 |
| | | | Nup93 | −1.7850 | 1.1110E-06 |
| Calcium signaling | | | Orc6 | −1.6260 | 1.5634E-04 |
| Npy | −3.2026 | 3.9755E-13 | Tmem144 | −1.6091 | 2.9040E-03 |
| Crem | −1.9204 | 2.4035E-09 | Vrk1 | −1.5578 | 1.0175E-08 |
| Gem | −1.8187 | 2.6277E-05 | Shmt1 | −1.5410 | 1.6394E-03 |
| Dusp1 | −1.5182 | 9.3142E-10 | Mcm3 | −1.5330 | 4.7341E-04 |
| Plat | −1.5137 | 8.2853E-03 | Plch1 | −1.5305 | 6.9677E-11 |
| Tpcn2 | −1.4906 | 3.5030E-03 | Hells | −1.5287 | 2.7324E-03 |
| Mif | −1.4597 | 9.0441E-04 | Mns1 | −1.5270 | 5.1788E-06 |
| Vcl | −1.3870 | 3.6093E-03 | Plat | −1.5137 | 8.2853E-03 |
| Serca2 | −1.2529 | 5.7379E-03 | Tubg1 | −1.5134 | 3.6446E-06 |
| | | | Dnmt1 | −1.4820 | 3.7757E-06 |
| ER sress | | | Junb | −1.4651 | 5.6951E-04 |
| Herpud1 | 1.8461 | 4.6638E-24 | Pcna | −1.4557 | 7.8393E-04 |
| Nucb1 | 1.4045 | 3.0361E-08 | Cast | −1.4562 | 9.4477E-04 |
| Hspa5 | 1.3517 | 4.1005E-04 | Net1 | −1.4507 | 1.6503E-03 |
| Dnajc3 | 1.3468 | 3.2649E-06 | Myo5a | −1.4252 | 1.2734E-03 |
| Ddit3 | 1.3153 | 7.5196E-03 | Alms1 | −1.4229 | 1.1049E-03 |
| Manf | 1.2009 | 3.2893E-02 | Chaf1a | −1.4137 | 6.1889E-03 |
| | | | Lig1 | −1.4101 | 9.1110E-04 |

*Table 1 continued on next page*

Table 1 continued

| Gene symbol | log2 Fold change | p value | Gene symbol | log2 Fold change | p value |
|---|---|---|---|---|---|
| Oxidative stress | | | Ramp2 | −1.3860 | 4.0237E-03 |
| Gstp1 | 1.6177 | 6.5804E-06 | Nphp4 | −1.3854 | 8.2551E-03 |
| Txnip | 1.5075 | 1.0152E-02 | Mcm6 | −1.3587 | 5.1079E-03 |
| Gstz1 | 1.4388 | 4.9681E-04 | Ywhah | −1.3561 | 7.3805E-05 |
| | | | Tubb4b | −1.3543 | 1.8881E-04 |
| Cell death | | | Rgs3 | −1.3424 | 3.2137E-04 |
| Card14 | 4.3450 | 8.5040E-32 | Bex2 | −1.3389 | 5.1152E-04 |
| Gdf15 | 3.1410 | 1.6785E-27 | Clic1 | −1.3291 | 3.9543E-04 |
| Bmp3 | 2.5051 | 5.7201E-18 | Polh | −1.3169 | 9.9122E-03 |
| Proc | 2.0628 | 1.5909E-07 | Tpm4 | −1.3125 | 3.3364E-04 |
| Rorc | 1.9612 | 7.4298E-07 | Uchl5 | −1.3085 | 8.1939E-03 |
| Bdnf | 1.8487 | 8.2780E-05 | Kpnb1 | −1.3081 | 3.7374E-05 |
| Herpud1 | 1.8461 | 4.6638E-24 | Phf6 | −1.3048 | 5.1520E-05 |
| Creb3l1 | 1.7775 | 5.6240E-11 | Pitpnm1 | −1.3032 | 5.6223E-04 |
| Eph7 | 1.7461 | 1.7641E-04 | Aim1 | −1.3023 | 9.1456E-03 |
| Pde3a | 1.7236 | 2.3400E-04 | Cdk5rap2 | −1.3022 | 1.7257E-03 |
| Ascl1 | 1.7219 | 6.1969E-04 | | | |
| Mpz | 1.7049 | 1.2768E-04 | Mitochondria and electron transport chain | | |
| Relt | 1.6992 | 3.9164E-07 | Ndufs2 | −1.6249 | 6.6049E-16 |
| Cnr1 | 1.6906 | 1.3432E-03 | Sdhc | −1.2008 | 9.8858E-04 |
| Osgin1 | 1.6732 | 1.0137E-06 | Cox6a2 | −5.1088 | 1.1287E-24 |
| Vip | 1.6721 | 1.5422E-03 | Cox6c | −1.2150 | 2.0212E-02 |
| Gstp1 | 1.6177 | 6.5804E-06 | Pdk1 | 2.0052 | 4.3578E-11 |
| Klf11 | 1.6138 | 6.0024E-05 | Pdk2 | 1.5897 | 1.6104E-04 |
| Rgn | 1.5784 | 4.3567E-03 | Pdk4 | 1.3010 | 6.7512E-02 |
| Dlc1 | 1.5682 | 9.9181E-05 | Pcx | −1.3978 | 4.8333E-09 |
| Rass2f | 1.5651 | 5.6005E-03 | Fh1 | −1.8589 | 1.7194E-07 |
| Wnt4 | 1.5573 | 4.4795E-08 | | | |
| Tle1 | 1.5488 | 1.0352E-13 | Non-beta cell hormones | | |
| Fgb | 1.5356 | 7.8501E-03 | Glucagon | 1.3727 | 7.0468E-03 |
| Bmpr1b | 1.5335 | 3.1844E-03 | Somatostatin | 1.1510 | 6.5084E-02 |
| Pycr1 | 1.5308 | 4.0752E-04 | Pancreatic polypeptide | 1.3546 | 1.9982E-03 |
| Cd44 | 1.5265 | 9.8606E-03 | Ghrelin | 1.2195 | 1.3036E-01 |
| Nod1 | 1.5259 | 5.4041E-05 | | | |
| Rasgrf2 | 1.5068 | 2.4891E-04 | | | |
| Dapk1 | 1.5036 | 7.5147E-06 | | | |

DOI: https://doi.org/10.7554/eLife.38472.013

## Mechanisms of impaired β-cell mass expansion in *Akita* neonates

Consistent with abrogated β-cell growth, the expression of proliferation and cell cycle genes was reduced in pre-weaning *Akita* neonates (*Table 1*). It has been reported that insulin-like growth factor 1 and 2 (IGF1 and IGF2) and epidermal growth factor (EGF) receptors are necessary for normal β-cell growth and differentiation (*Kulkarni et al., 2002*; *Miettinen et al., 2008*). The expression of EGF, IGF1 and IGF2 receptors was indeed decreased in *Akita* islets, whereas the expression of the insulin receptor remained unchanged (*Table 1*). Growth factors mediate their effects via IRS proteins with

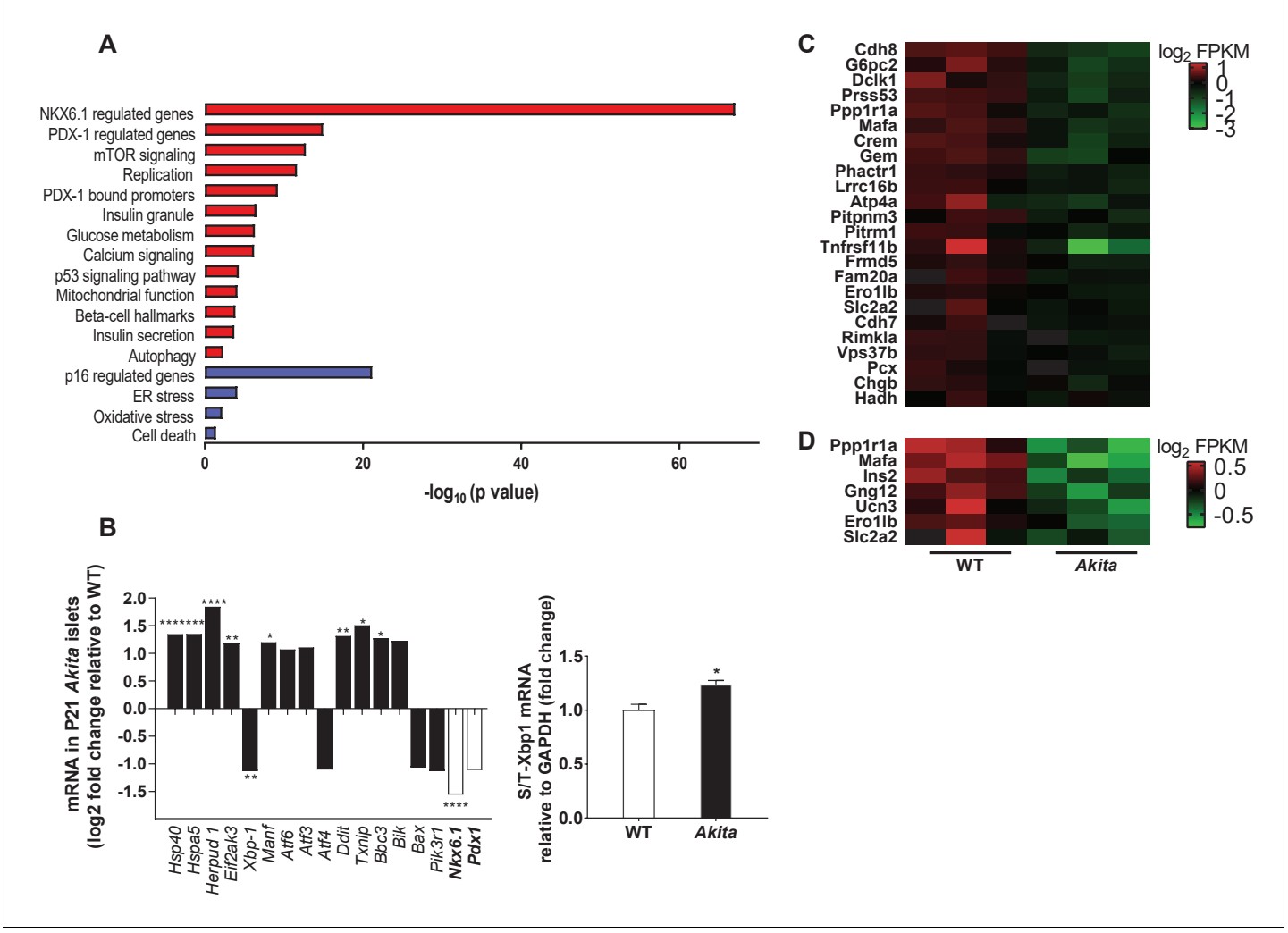

**Figure 5.** Transcriptomic analysis of ER stress markers and β-cell gene signature in neonate *Akita* islets. (a) RNA-seq comparing the transcriptome of islets from P19-21 *Akita* and age-matched control mice (n = 3 samples in each group, each sample is a pool of islets from three mice). Columns represent pathways that are differentially regulated in *Akita* mice; (b) expression of UPR and apoptosis genes and of *Nkx6.1* and *Pdx1* in islets of *Akita* compared to control mice at P19-21. Spliced and total *Xbp1* were also quantified by qPCR. The spliced/total *Xbp1* ratio is shown beside (n = 3); (c–d) heat map of genes regulated by NKX6.1 (c) and PDX-1 (d) in *Akita* islets and controls. *p<0.05, **p<0.01, ***p<0.001, ****p<0.0001.

DOI: https://doi.org/10.7554/eLife.38472.010

The following figure supplement is available for figure 5:

**Figure supplement 1.** Islet composition of wild-type and *Akita* mice.

DOI: https://doi.org/10.7554/eLife.38472.011

subsequent activation of PI3 kinase and its downstream target AKT. The expression of *Pik3r1* encoding for the regulatory unit of PI3 kinase (P85α) was decreased (*Figure 5b*), along with marked inhibition of AKT activity (*Figure 7a*). mTORC1 is a protein complex that integrates signals from nutrients, growth factors, hormones and stress to regulate cell growth and proliferation, which is indispensable for embryonic and postnatal β-cell growth and maturation (*Ni et al., 2017*). Western blotting showed that also mTORC1 activity was markedly inhibited in neonatal *Akita* islets, evident by decreased protein levels and Ser240/244 phosphorylation of ribosomal S6 (*Figure 8a*). Eukaryotic translation initiation factor 4E binding protein (4E-BP1) dephosphorylation was reflected in the shift from the highly phosphorylated γ-band to the nonphosphorylated β-band as previously described (*Ni et al., 2017*) (*Figure 8a*). Immunostaining showed that the number of phospho-S6+ β-cells was high in newborn β-cells and decreased over time (*Figure 8b*). On the contrary, mTORC1 activity (S6

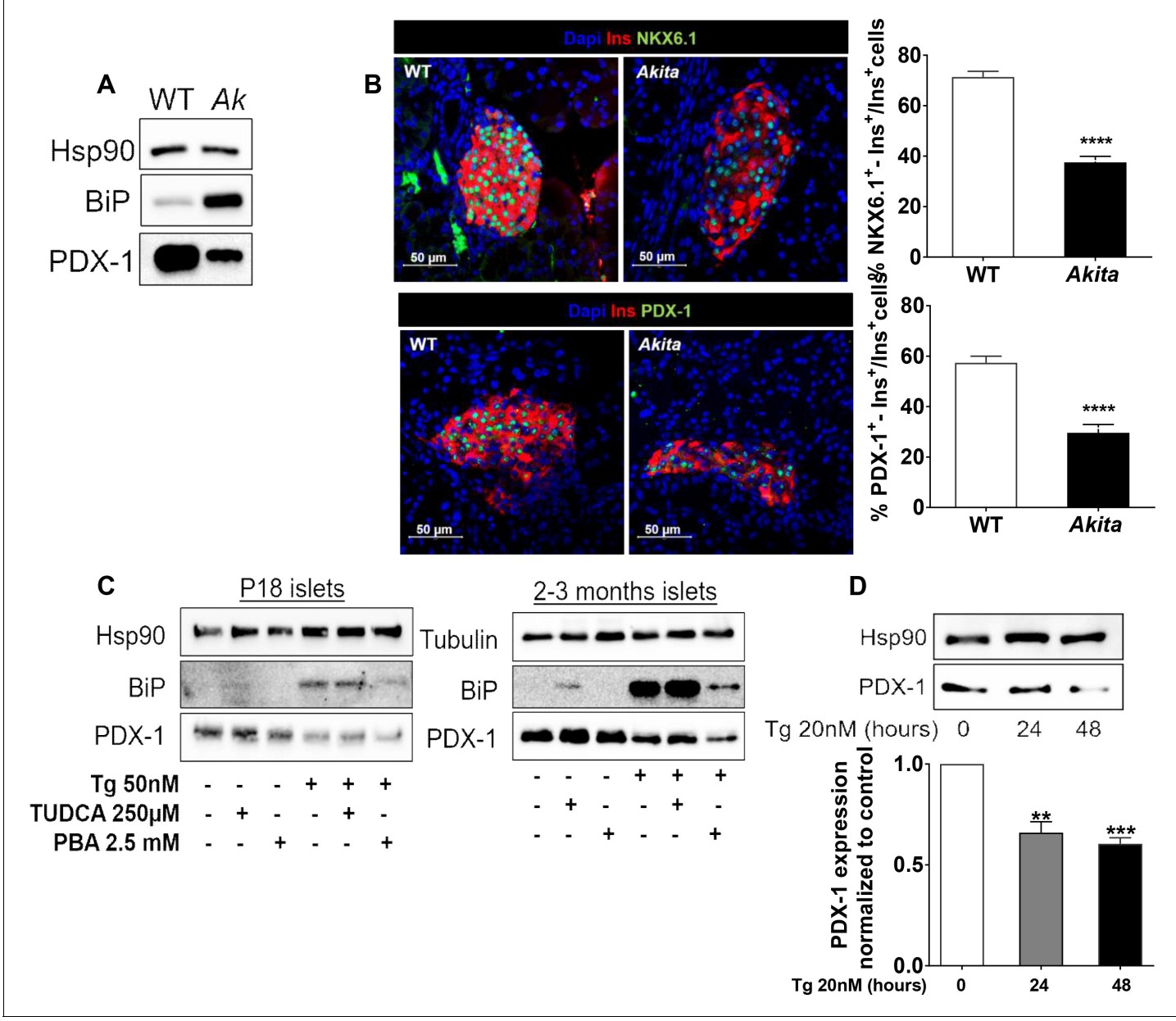

**Figure 6.** Effects of ER stress on the expression of β-cell transcription factors in neonate *Akita* islets (P19-21) and islets treated with thapsigargin. (a) PDX-1 and BiP protein level analyzed by Western blotting (n = 3, each sample is a pool of islets from four to six mice); (b) quantification of NKX6.1 (n = 3 mice in each group; 1646 WT and 728 *Akita* β−cells), and PDX-1 (n = 3 mice in each group; 1534 WT and 844 *Akita* β−cells) expressing β-cells. Pancreatic sections were immunostained for NKX6.1 or PDX-1 and insulin. The percentage of NKX6.1- and PDX-1-positive β-cells is shown. (c) Islets from young (P19-21) and adult wild-type mice were treated with low-dose thapsigargin (50 nmol/l) and TUDCA (250 µmol/l) or PBA (2.5 mmol/l) for 48 hr with daily media changes and further analyzed by western blotting for PDX-1 and BiP (n = 3, each sample is a pool of islets from six to nine mice); (d) INS-1E cells were treated with 20 nmol/l thapsigargin for 24 and 48 hr followed by western blotting for PDX-1. **p<0.01, ***p<0.001, ****p<0.0001.
DOI: https://doi.org/10.7554/eLife.38472.012

phosphorylation) in the exocrine pancreas was low during the neonatal period and was markedly enhanced in adult mice (*Figure 8b*). The number of phospho-S6$^+$ β-cells was lower in *Akita* mice than in controls already at P1-2 and at P19-21, further indicating that mTORC1 activity was decreased in neonate *Akita* β-cells (*Figure 8b*). In adult islets the number of S6$^+$ β-cells was small and mTORC1 activity was increased in *Akita*, despite sustained inhibition of AKT signaling (*Figure 8a and 7a*). We and others have previously shown that in diabetes hyperglycemia stimulates

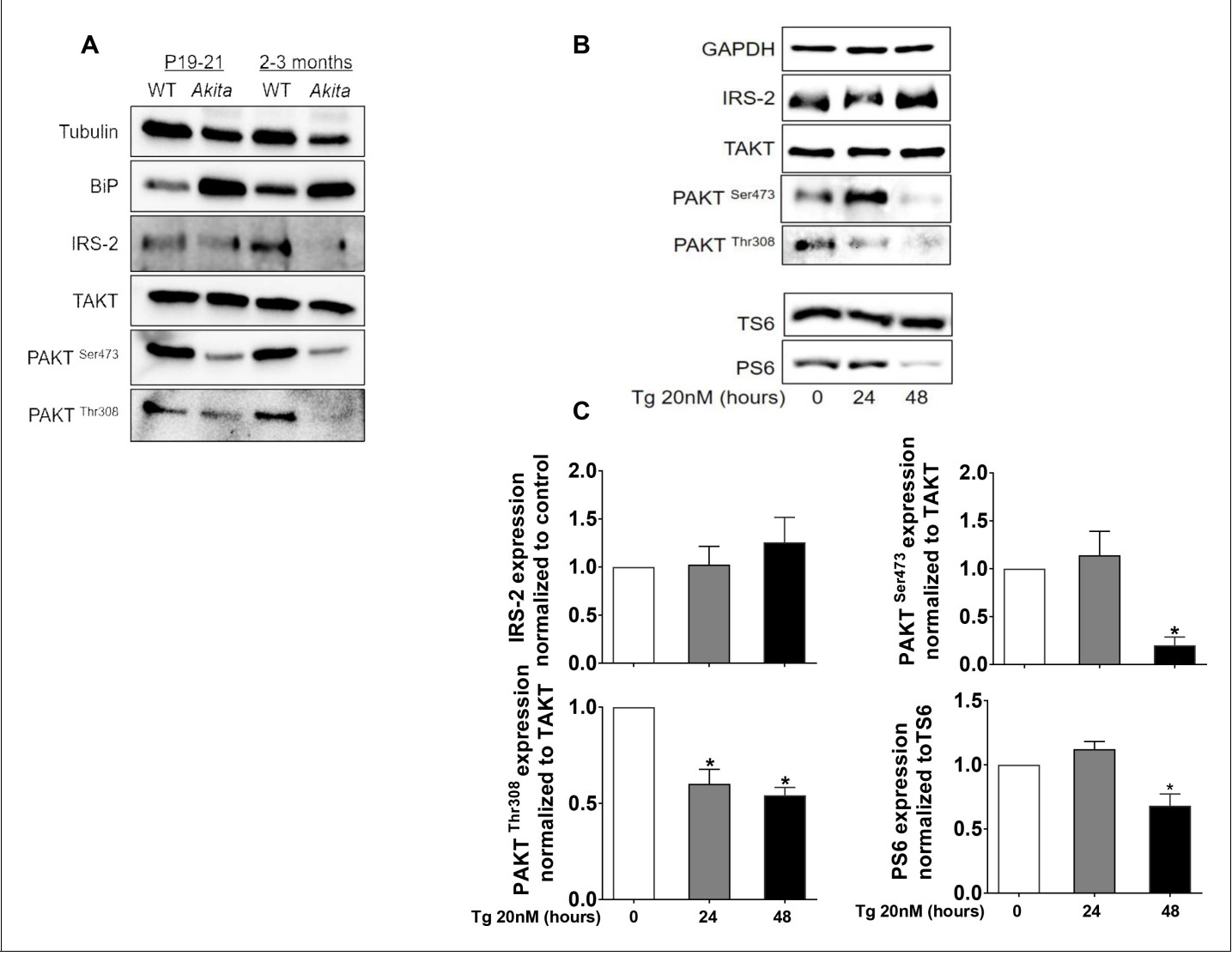

**Figure 7.** Effects of ER stress on IRS2/Akt signaling in *Akita* islets and in INS-1E treated with low-dose thapsigargin. (a) IRS2/Akt signaling in islets from neonate (P19-21) and adult wild-type and *Akita* mice. Each sample is a pool of islets from 4 to 15 mice (n = 4 for neonate islets and n = 2 for adult islets). (b–c) INS-1E cells were treated with 20 nmol/l thapsigargin for 24 and 48 hr followed by western blotting for IRS2, total and phosphorylated Akt (Ser473 and Thr308) and S6 (Ser240/244). A representative experiment (b) and quantification (c) are shown (n = 4–6). *p<0.05.
DOI: https://doi.org/10.7554/eLife.38472.014

mTORC1 activity (*Fraenkel et al., 2008*; *Yuan et al., 2017*). Treatment of adult *Akita* mice with the glucosuric drug dapagliflozin for 72 hr decreased blood glucose and abrogated S6 phosphorylation (*Figure 8c*), indicating that mTORC1 activation in diabetic *Akita* β-cells is mediated via hyperglycemia. Consistent with the findings in neonate *Akita* islets, treatment of INS-1E cells with low-dose thapsigargin for 48 hr did not affect IRS2 protein level and inhibited AKT and S6 phosphorylation (*Figure 7b–c*), suggesting that ER stress inhibits AKT-mTORC1 signaling. We next studied whether treatment with chemical chaperones can prevent the downregulation of mTORC1 and increase β-cell proliferation in *Akita* neonates. Intriguingly, both TUDCA and PBA further decreased mTORC1 activity in *Akita* β-cells (*Figure 8—figure supplement 1a*). In vivo, treatment of *Akita* neonates with TUDCA for 48 hr decreased β-cell proliferation (*Figure 8—figure supplement 1b*).

In summary, in *Akita* islets mTORC1 is inhibited during the neonatal period in parallel to the β-cell growth arrest. Treatment with chemical chaperones failed to correct the early β-cell growth arrest.

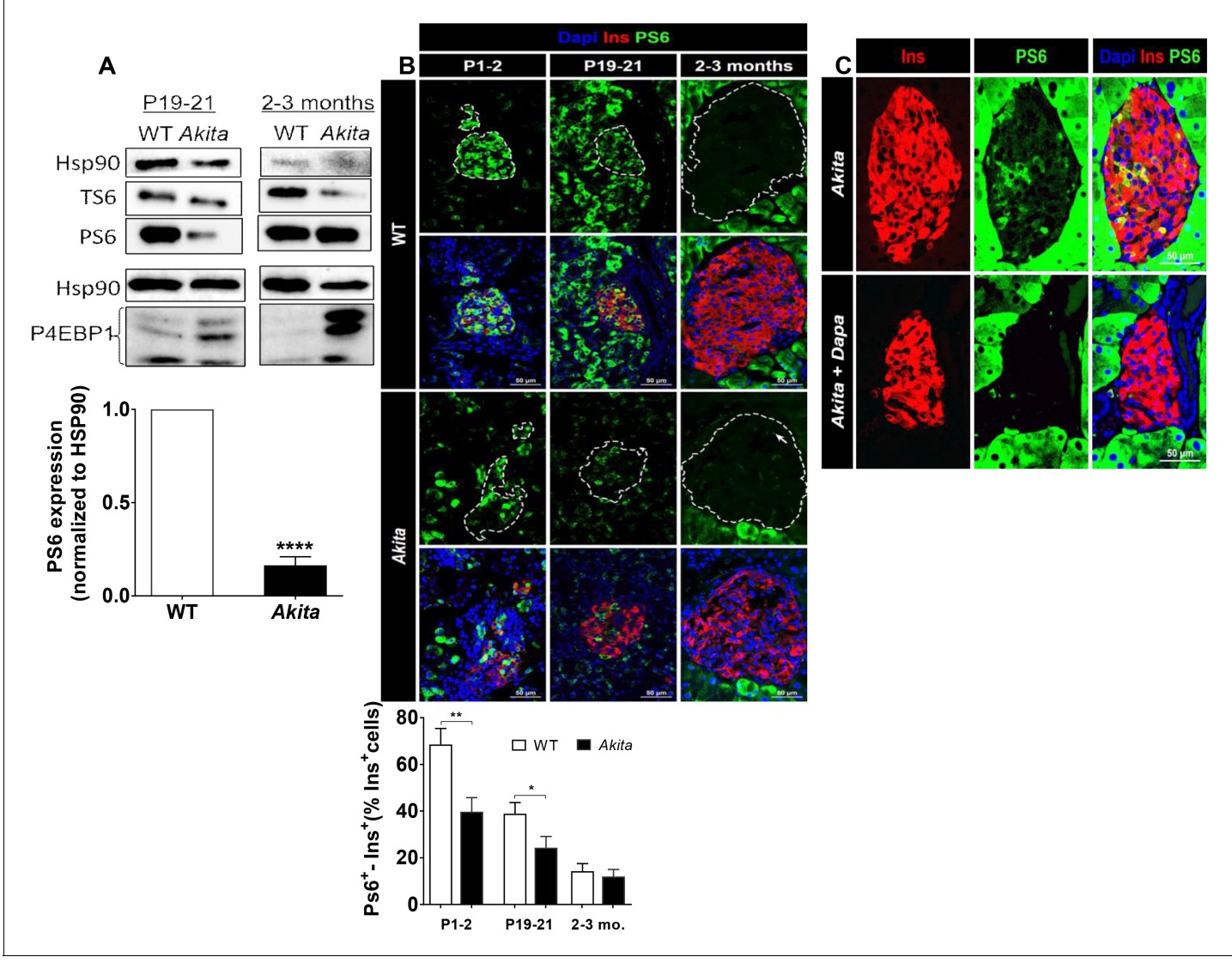

**Figure 8.** mTORC1 signaling in neonate and adult *Akita* islets. (a) Western blot analysis of S6 and 4EBP1 phosphorylation in islets of neonate (P19-21) and adult wild-type and *Akita* mice. Quantification of phosphorylated S6 in neonate *Akita* compared to control islets is shown (n = 3, each sample is a pool of islets from 4 to 7 mice); (b) immunostaining for phospho-S6 on pancreatic sections of P1-2, P19-21 and adult *Akita* mice and age-matched controls and quantifications of the percentage of S6$^+$ β-cells (P1-2: n = 4 mice in each group; 1159 WT and 1655 *Akita* β−cells; P19-21: n = 6 mice in each group; 2259 WT and 1567 *Akita* β−cells; adult: n = 4–5 mice in each group; 2391 WT and 1383 *Akita* β-cells). Islet boundaries are marked by dotted line; (c) adult *Akita* mice were treated with 25 mg/kg dapagliflozin in drinking water for 72 hr. Blood glucose in dapagliflozin-treated *Akita* mice was ~ 200 mg/dl compared to ~ 500 mg/dl in control *Akita* mice. Pancreatic sections were immunostained for insulin and phospho-S6 (n = 3 mice in each group). *p<0.05, **p<0.01, ****p<0.0001.

DOI: https://doi.org/10.7554/eLife.38472.015

The following figure supplement is available for figure 8:

**Figure supplement 1.** Effects of chemical chaperones on mTORC1 activity in neonate *Akita* islets and controls.

DOI: https://doi.org/10.7554/eLife.38472.016

## Restoration of mTORC1 activity improves β-cell function and diabetes in *Akita* mice

The TSC1/TSC2 complex is a key negative upstream regulator of mTORC1. Constitutive activation of mTORC1 by *Tsc2* knockout in β-cells modulates β-cell mass in a biphasic manner (*Bartolomé et al., 2014*; *Shigeyama et al., 2008*). In young mice, constitutive mTORC1 activation increases β-cell number and size, whereas in old mice the animals develop diabetes due to increased

β-cell apoptosis. Because ER stress inhibited mTORC1 and β-cell growth in neonates, we studied whether stimulation of mTORC1 could rescue diabetes in *Akita* mice. We generated heterozygous and homozygous β*Tsc1* knockout *Akita* mice (*RIP-Cre:Tsc1*$^{flox/+}$*:Akita* and *RIP-Cre:Tsc1*$^{flox/flox}$*:Akita* mice) by crossing *Akita* mice with *Rosa-26-floxed Tsc1* mice and with *RIP-Cre* mice. β*Tsc1*$^{+/+}$, β*Tsc1*$^{+/-}$ and β*Tsc1*$^{-/-}$ mice were used as controls. It has been previously reported that the *RIP-Cre* alone without recombination at *lox* sites is associated with glucose intolerance and even frank diabetes (*Lee et al., 2006*). We found that in wild-type mice, expression of *RIP-Cre* induced only modest glucose intolerance even in adult mice (*Figure 9—figure supplement 1a*). Moreover, it did not affect fed blood glucose either in wild-type or in *Akita* mice (*Figure 9—figure supplement 1b*), and the insulin sensitivity of *Akita* mice was unaltered (*Figure 9—figure supplement 1c*). We therefore believe that this is a valid model to test the effects of mTORC1 activation on diabetes and β-cell function in *Akita* mice.

We first studied the effect of *Tsc1* knockout on mTORC1 activity in neonates at P19-21. In *Akita* neonates TSC1 deficiency increased mTORC1 activity compared to *Akita* controls, evident by S6 phosphorylation (*Figure 9a*). Activation of mTORC1 did not affect the expression of BiP (*Figure 9b*), suggesting that this did not have a major effect on β-cell ER stress. TSC1 deficiency in *Akita* mice increased β-cell size (*Figure 9c*) and proliferation (*Figure 9d*). At P30-35, mTORC1 activation did not affect β-cell proliferation either in heterozygous or homozygous *Tsc1* knockout mice (*Figure 9d*), indicating that stimulation of mTORC1 induced β-cell proliferation only in young mice prior to weaning.

We then tested the effects of β-cell TSC1 deficiency on the metabolic state of *Akita* mice after weaning. IPGTT performed at the age of 3–4 weeks showed that glucose tolerance was improved or normalized in β*Tsc1*$^{+/-}$ and β*Tsc1*$^{-/-}$*Akita* mice (*Figure 10a–b*). TSC1 deficiency doubled pancreatic insulin content in control mice and increased it fivefold in *Akita* mice (*Figure 10c–d*). Islet insulin content of *Akita* mice crossed with the *Tsc1* null was twofold higher compared to *Akita* islets (*Figure 10e*). Glucose-stimulated insulin secretion remained markedly reduced in vivo and ex vivo (*Figure 10f–g*), indicating that stimulation of mTORC1 improved the metabolic state by increasing β-cell mass and islet insulin content without affecting the fundamental defects in the insulin response to stimulus. Intriguingly, activation of mTORC1 in pre-diabetic *Akita* islets did not affect PDX-1 and NKX6.1 expression (*Figure 10—figure supplement 1a–d*). Collectively, these findings indicate that activation of mTORC1 improved glycemia by increasing β-cell mass and islet insulin content despite persistent ER-stress-induced β-cell dysfunction. We followed a small number of *Akita* mice with restored mTORC1 for 3 months. Part of the mice became mildly hyperglycemic, whereas others developed overt diabetes with severe hyperglycemia (*Figure 10—figure supplement 2*). Thus, life-long ER stress might eventually lead to diabetes despite the initial increase in β-cell mass with heterogeneity in the timing of appearance and severity of hyperglycemia.

## Discussion

We found that the ER stress of neonate β-cells interrupted their proliferation and cell size growth, resulting in low β-cell mass accompanied by severe insulin deficiency. The decline in β-cell proliferation along with attenuation of β-cell hypertrophy caused a ~ 70% decrease in β-cell mass together with marked depletion of islet insulin content and blunt insulin response to glucose. These deficiencies culminated in diabetes when nutrient load was increased after weaning. β-Cell growth arrest in *Akita* neonates was associated with transient inhibition of mTORC1. Interestingly, in adult *Akita* mice mTORC1 activity was increased, most probably due to hyperglycemia, and β-cell growth was resumed, albeit without 'catch-up' growth, hence β-cell mass remained reduced. Importantly, partial restoration of mTORC1 activity in neonate *Akita* β-cells was sufficient to rescue β-cell expansion with marked improvement of glucose tolerance despite ongoing ER stress and β-cell dysfunction, indicating that mTORC1 inhibition plays a key role in the pathophysiology of neonatal diabetes (*Figure 10h*).

Our findings highlight the importance of postnatal β-cell growth and differentiation for normal glucose homeostasis in adult life. During the neonatal period, β-cells expand rapidly by proliferation, followed by hypertrophy after the transition from suckling to regular chow. These dynamics of β-cell number and size culminate in ~ 8-fold increase of β-cell mass in young adult animals, which seems to be a *sine qua non* condition for coping with the increased insulin demand of adult life. In our model

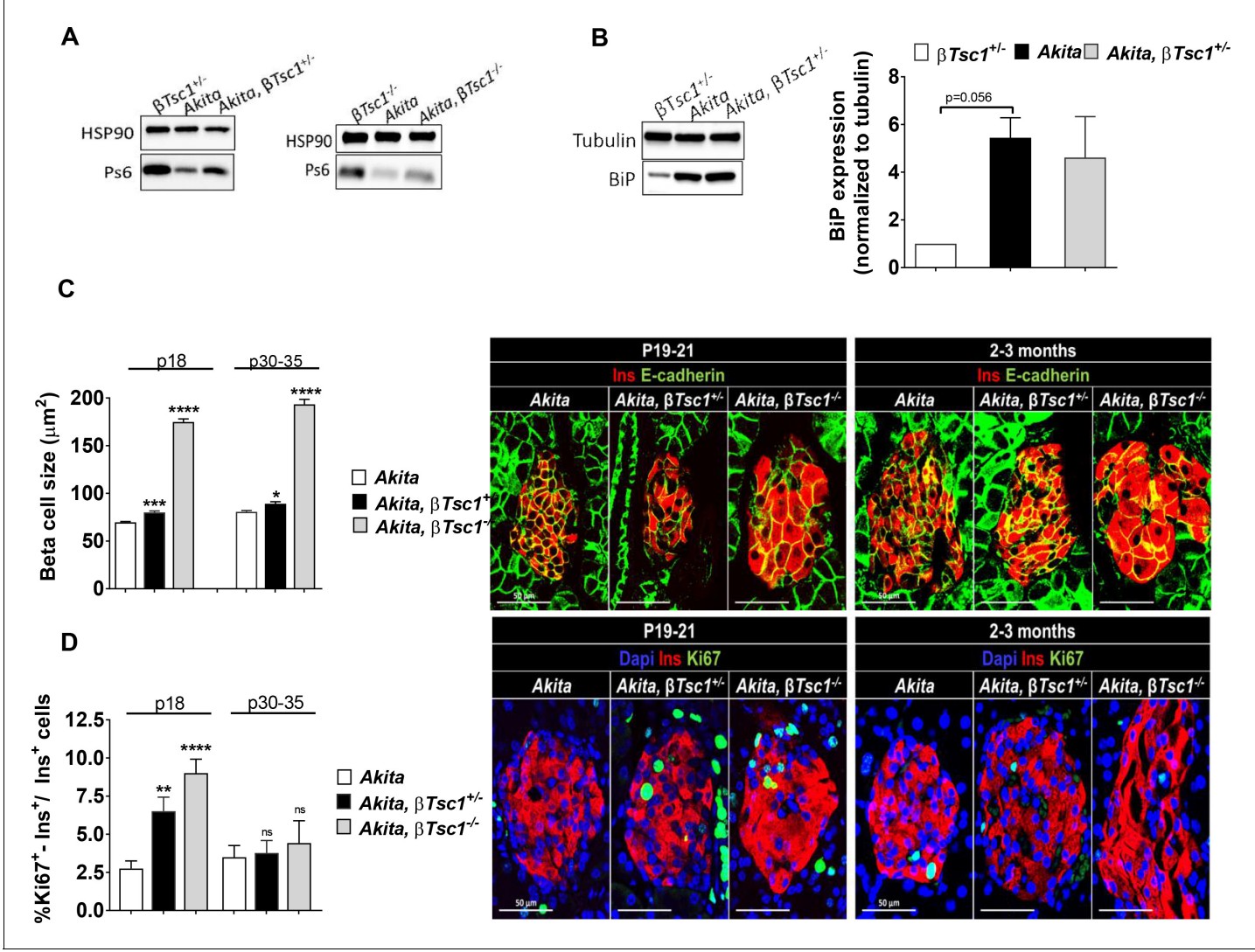

**Figure 9.** Effects of mTORC1 activation in neonate *Akita* β-cells on β-cell size and proliferation. Studies were performed on heterozygous and homozygous βTsc1 knockout *Akita* mice (*RIP-Cre:Tsc1*^flox/+^:*Akita* (*Akita*, β*Tsc1*^+/-^) and *RIP-Cre:Tsc1*^flox/flox^:*Akita* (*Akita*, β*Tsc1*^-/-^). *Tsc1*^flox/+^:*Akita* and *Tsc1*^flox/flox^:*Akita* were used as *Akita* controls. *RIP-Cre:Tsc1*^flox/+^ mice (β*Tsc1*^+/-^) and *RIP-Cre:Tsc1*^flox/flox^ mice (β*Tsc1*^-/-^) were used as WT controls (a, b). (a) Western blotting for phospho-S6 on islets from homozygous and heterozygous knockout mice and matched controls (n = 4, each sample is a pool of islets from two to four mice); (b) Western blotting and quantification of BiP expression in wild-type, *Akita* and *Akita*, β*Tsc1*^+/-^ mice (n = 4, each sample is a pool of islets from two to four mice); (c) β-cell size was assessed by immunostaining for insulin and E-cadherin (n = 400–500 β-cells per group), (d) β-cell proliferation was assessed by immunostaining for insulin and Ki67 (n = 1200–1400 β−cells per group). Quantifications and representative images are shown. *p<0.05, **p<0.01, ***p<0.001, ****p<0.0001.

DOI: https://doi.org/10.7554/eLife.38472.017

The following figure supplement is available for figure 9:

**Figure supplement 1.** Metabolic characterization of *RIP-Cre* mouse.

DOI: https://doi.org/10.7554/eLife.38472.018

(*Akita* mice) ER stress is induced by the expression of a mutant, unfoldable insulin which creates protein aggregates in the ER. Insulin is expressed at day E11.5 and therefore some degree of ER stress is expected to occur already in fetal *Akita* β-cells. Nevertheless, neonate *Akita* mice had normal β-cell mass. Also affected human subjects with *Akita* diabetes are born with normal body weight and are normoglycemic at birth (see accompanying paper by Balboa *et al*), indicating that β-cell dysfunction develops after birth.

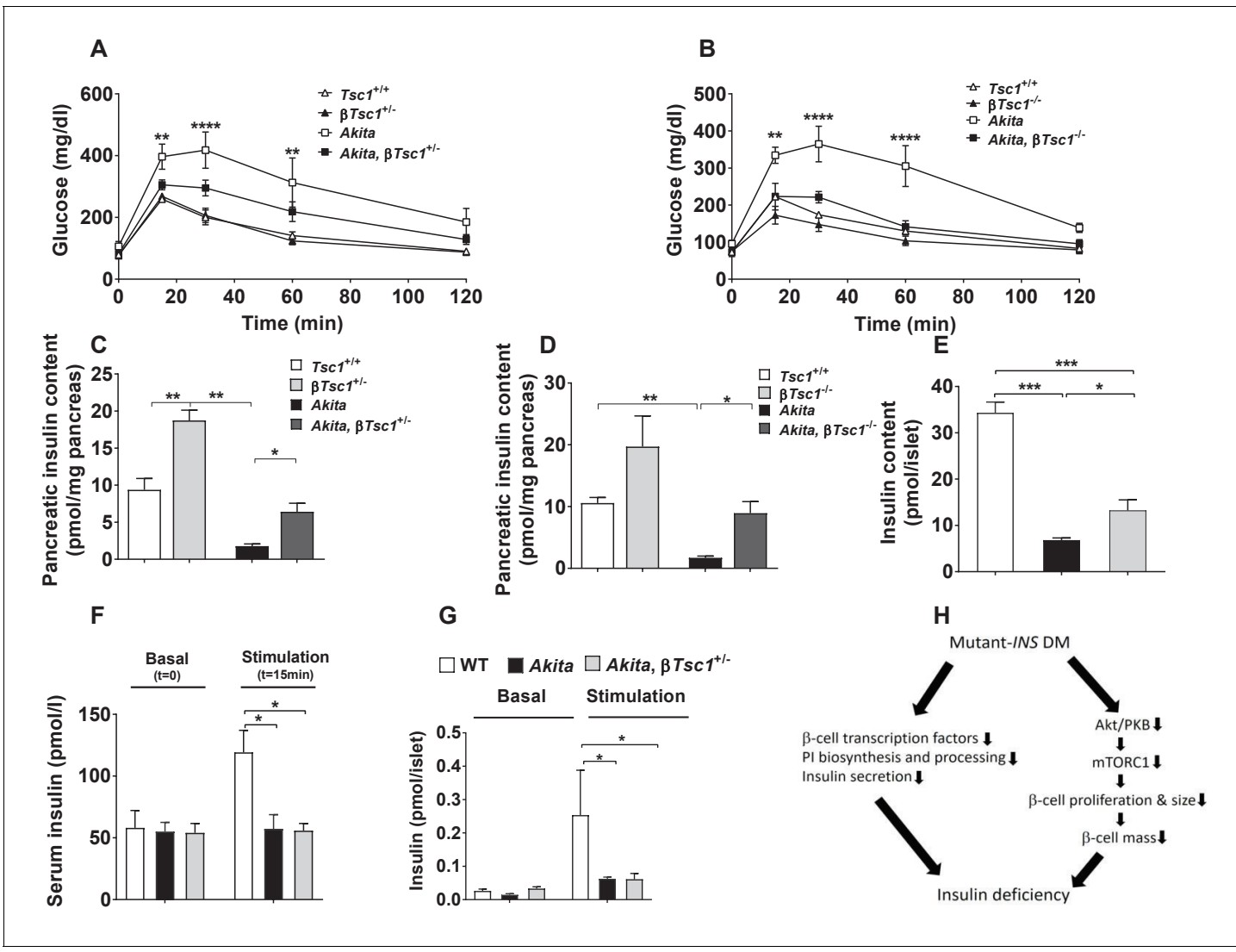

**Figure 10.** Effects of mTORC1 activation in neonate *Akita* β-cells on diabetes. (a–b) IPGTT at P30-35: glucose (1 g/kg) was injected IP after an overnight fast; (a) heterozygous *Tsc1* knockout *Akita* mice (*RIP-Cre:Tsc1^flox/+^:Akita* (*Akita, βTsc1^+/-^*) and matched controls: *Tsc1^flox/+^* mice (*Tsc1^+/+^*), *RIP-Cre: Tsc1^flox/+^* mice (*βTsc1^+/-^*), and *Tsc1^flox/+^:Akita* (*Akita*) (n = 3–5 mice in each group); (b) homozygous *Tsc1* knockout *Akita* mice (*RIP-Cre:Tsc1^flox/flox^:Akita* (*Akita, βTsc1^-/-^*) and matched controls: *Tsc1^flox/flox^* mice (*Tsc^+/+^*), *RIP-Cre:Tsc1^flox/flox^* mice (*βTsc1^-/-^*), and *Tsc1^flox/flox^:Akita* (*Akita*) (n = 3–5 in each group); (c–d) pancreatic insulin content of heterozygous and homozygous *Tsc1* knockout *Akita* mice and matched controls at P30-35 (WT (n = 7), *Akita* (n = 11), *Akita, βTsc1^+/-^* (n = 3) and *Akita, βTsc1^-/-^* (n = 4); (e) islet insulin content. (f–g) Effects of mTORC1 activation in neonate *Akita* β-cells on insulin secretion in vivo and ex vivo. (f) insulin secretion in response to IP glucose injection (n = 6 mice in each group); (g) islets were isolated from *Tsc1^flox/+^* WT mice (WT), *Tsc1^flox/+^:Akita* (*Akita*) and *RIP-Cre:Tsc1^flox/+^:Akita* (*Akita, βTsc1^+/-^*) mice and insulin secretion assessed following static incubations at basal (3.3 mmol/l) and stimulated (16.7) mmol/l glucose. (h) a model of the pathophysiology of permanent neonatal diabetes. *p<0.05 **, p<0.01, ***, p<0.001****, p<0.0001.

DOI: https://doi.org/10.7554/eLife.38472.019

The following figure supplements are available for figure 10:

**Figure supplement 1.** Effects of mTORC1 activation in *Akita*β-cells on PDX-1 (a, b) and NKX6.1 expression (c, d).
DOI: https://doi.org/10.7554/eLife.38472.020
**Figure supplement 2.** Fed blood glucose of *Tsc1^flox/+^* mice (WT), *Tsc1^flox/+^:Akita* (*Akita*) and heterozygous *Tsc1* knockout *RIP-Cre:Tsc1^flox/+^:Akita* (*Akita,βTsc1^+/-^*) mice at the age of 2–3 months.
DOI: https://doi.org/10.7554/eLife.38472.021

Several lines of evidence indicate that fetal and neonate β-cells respond to stress by slowing replication. A striking example is intrauterine growth retardation (IUGR), where placental insufficiency generates hypoxia and nutrient deprivation, resulting in decreased β-cell proliferation and mass *in utero* (*Thompson et al., 2010*). Infants with IUGR exhibit impaired insulin secretion and show a high incidence of T2D in adulthood (*Barker, 2006*). Similarly, malnutrition and low-protein diet during pregnancy restrict the number of β-cells in the fetal pancreas (*Alejandro et al., 2014*; *Dumortier et al., 2011*; *Garofano et al., 1998*). The Wolcott-Rallison syndrome results from mutations in PERK (EIF2AK3), leading to permanent neonatal diabetes due to β-cell ER stress. Similar to *Akita* mice, PERK-deficient mice exhibited severe defects in neonatal β-cell proliferation, resulting in low β-cell mass and β-cell dysfunction (*Zhang et al., 2006*). Most importantly, in the accompanying paper Balboa *et al* show that the proliferation rate of β-like cells derived from induced-pluripotent stem (iPS) cells from human subjects carrying missense *INS* mutations, which disrupt the proinsulin inter-chain disulphide bonds formation similar to the *Akita* mutation, was reduced compared to control cells in which the mutation was corrected by CRISPR. These findings strongly suggest that the proliferation inhibitory response to ER stress is a general phenomenon, and indeed relevant to disease pathophysiology in man.

We performed an unbiased transcriptomic analysis in *Akita* neonates and studied insulin/IGF-1 signaling to clarify how ER stress induces β-cell growth arrest. Strikingly, we found that the genetic program that governs β-cell growth, including growth factor receptors (IGF-1R, IGF-2R and EGFR) and other replication genes, was downregulated. Moreover, AKT-mTORC1 signaling was vigorously suppressed. Previous reports showed that ER stress leads to suppression of insulin receptor signaling in peripheral tissues through hyperactivation of c-Jun N-terminal kinase (JNK) and subsequent serine phosphorylation of insulin receptor substrate-1 (IRS-1) (*Ozcan et al., 2004*). It has been recently shown that growth factor receptor bound protein 10 (GRB10), a key negative regulator of insulin, IGF1 and mTORC1 signaling is activated by ER stress via an ATF4-mediated increase in *Grb10* transcription (*Luo et al., 2018*). Interestingly, GRB10 has been implicated in the regulation of β-cell proliferation and function (*Zhang et al., 2012*; *Prokopenko et al., 2014*). We found that in neonate *Akita* islets, the expression of the regulatory unit of PI3 kinase, *Pik3r1* (*p85α*), which is essential for PI3 kinase activation by growth factors is decreased. P85α directly interacts with sXbp1 and mediates its transport to the nucleus (*Winnay et al., 2014*; *Park et al., 2010*), hence P85α may have a dual role in the regulation of insulin/IGF1-1 and UPR signaling in response to ER stress. Collectively, multiple mechanisms might be involved in the inhibition of insulin/IGF-1 signaling by ER stress. Of note, mTORC1 activity was also decreased in β-like cells derived from iPS cells from human subjects carrying the *INS* C96R (*Akita*) mutation (accompanying paper). Accumulating data suggest that mTORC1 is a master regulator of β-cell growth during early development. mTORC1 inhibition by β-cell-specific deletion of *Raptor* disrupts mitochondrial function, and postnatal β-cell growth and functional maturation (*Ni et al., 2017*), thus mimicking our findings in neonate *Akita* islets. S6K1 deficiency in mice results in IUGR and impairment of β-cell growth in utero (*Um et al., 2015*). Feeding pregnant mice with a low protein diet decreased β-cell proliferation, mass and function in the offspring in an mTORC1-dependent manner (*Alejandro et al., 2014*). Consistent with this ubiquitous role of mTORC1 in regulating cell size and proliferation, decreased mTORC1 activity in *Akita* neonate islets was associated with β-cell growth arrest.

Others and we have previously shown that mTORC1 promotes ER stress, and its inhibition may prevent apoptosis under ER stress conditions (*Yuan et al., 2017*; *Bachar et al., 2009*; *Bachar-Wikstrom et al., 2013*; *Guha et al., 2017*). Therefore, mTORC1 down-regulation in neonate *Akita* islets can be viewed as an adaptive response aimed to alleviate ER stress and promote β-cell survival by halting anabolic, energy consuming processes. However, during early stages of development this adaptive mechanism is counter-productive, resulting in marked impairment of β-cell expansion, and consequently leads to future development of diabetes. Therefore, mTORC1 may be viewed as a double-edged sword in the context of β-cell ER stress: on one hand mTORC1 activation may promote ER stress, on the other hand its inhibition early in life impairs β-cell growth and differentiation. Interestingly, it has been recently suggested that mature tissues universally respond to cellular injury by first shutting down mTORC1, followed by its reactivation which is required for cell cycle entry and tissue repair; this process was termed paligenosis (*Willet et al., 2018*). In *Akita* mice, reactivation of mTORC1 occurred after weaning and the development of hyperglycemia when the β-cells already lost their ability to proliferate, resulting in permanent β-cell deficiency.

Differentiation of *Akita* β-cells was impaired, evident by decreased expression of genes regulating β-cell identity and function. These changes preceded the development of overt diabetes and are most likely secondary to ER stress per se; this is in contrast with the common view that β-cell dedifferentiation in diabetes is secondary to chronic hyperglycemia (*Wang et al., 2014*). β-Cell transcription factors, including PDX-1 and NKX6.1 and their downstream target genes, were decreased in normoglycemic *Akita* neonates. PDX-1 and NKX6.1 instruct β-cell differentiation during development, but are also essential for maintaining β-cell identity and function in adult animals. The latter has also been implicated in the regulation of β-cell proliferation (*Tessem et al., 2014*). Restoration of mTORC1 activity in *Akita* β-cells increased β-cell expansion and ameliorated diabetes without increasing PDX-1 and NKX6.1 expression and glucose-stimulated insulin secretion, further indicating that β-cell growth arrest induced by mTORC1 inhibition plays a key role in the pathophysiology of permanent postnatal diabetes.

Our findings have implications not only for the pathophysiology of rare monogenic forms of diabetes, but also for T2D. There is extreme heterogeneity in β-cell mass in healthy individuals as well as subjects with T1D and T2D, which is poorly understood (*Campbell-Thompson et al., 2016*; *Cigliola et al., 2016*). Adult β-cell mass is likely a key factor in the risk of developing T2D in the context of obesity and insulin resistance. Since β-cell proliferation is low in the adult, especially in humans, it is generally accepted that impaired β-cell proliferation plays a minor role in the pathophysiology of diabetes. However, genome-wide association studies do point to the importance of β-cell proliferation as a determinant of T2D (*Thomsen et al., 2016*). β-Cell expansion during the fetal and early neonatal period is extensive, and therefore impairment of β-cell proliferation during these early developmental stages will strongly impact the ultimate β-cell mass and function. Our data, although emanating from a neonatal diabetes model can be relevant also to other forms of diabetes, for example T2D in which interplay of genetics (variants in cell cycle genes) and environmental factors like viral infections, nutritional stressors or noxious chemicals during the early postnatal period, may induce silent but detrimental effects on β-cell mass via the ER stress-mTOR pathway, predisposing to diabetes in adulthood.

## Materials and methods

### Animals

Mouse strains used included *RIP-Cre* (*Gannon et al., 2000*), *Rosa26-LSL-Yfp* (*Srinivas et al., 2001*), *Akita* (*Ins2*^WT/C96Y^) (The Jackson Laboratory), *Tsc1*^fl/fl^ (a kind gift from Dr. B. Tirosh, The Hebrew University, Jerusalem). The genetic background of the *Tsc1*^fl/fl^ mice is 129S4/SvJae strain. *Ins2*^C96Y^ *Akita* and the *RIP-Cre* mice were generated on the background of C57BL/6J mice. The *Rosa26-LSL-Yfp* mice are a mixture of the 129 × 1/SvJ and of C57BL/6J as previously described (*Srinivas et al., 2001*). *Akita* males were selectively chosen for all analyses, since they develop a more severe form of diabetes compared to females. Mice were housed under similar conditions with 12 hr light/dark cycles with free access to food and water at The Hebrew University animal care unit.

### Metabolic assays

For assessment of glucose tolerance, mice fasted for 16 hr or 4 hr were given 1.0 or 1.5 g/kg glucose IP followed by consecutive blood glucose measurements. Tail blood glucose was monitored using an Accuchek glucometer (Roche Diagnostics GmbH, Mannheim, Germany). For measurement of serum insulin blood samples were collected either from the tail or from the facial vein using heparin coated capillaries or tubes at the start and 15 min after glucose injection. Plasma samples were analyzed using ultrasensitive insulin kits (Mercodia, Uppsala, Sweden and Crystal Chem Inc., IL). Pancreatic insulin content was analyzed in whole pancreas extracts. Pancreases were isolated, homogenized and insulin was extracted overnight in acid ethanol at 4°C. Insulin content was determined by an ELISA kit (Mercodia). Animal use was approved by the Institutional Animal Care and Use Committee of the Hebrew University.

### Islet isolation, β-cell line culture, and experimental protocols

The rat insulinoma cell line INS-1E was kindly provided by Prof. M. Walker (The Weizmann Institute of Science, Rehovot, Israel) and grown as previously described (*Luo et al., 2018*). Mycoplasma

contamination was examined periodically and the tests showed no evidence for contamination. Functionality of the cell line was validated by checking periodically their glucose stimulated insulin secretion. Islets were isolated by ductal perfusion of collagenase P (Roche). Hand-picked islets were plated for overnight recovery in RPMI-1640 medium containing 11.1 mmol/l glucose (Biological Industries) supplemented with 10% FBS, L-glutamine and penicillin-streptomycin in a 37°C, 5% $CO_2$ incubator before experimental procedures. For static glucose-stimulated insulin secretion tests, batches of 25 islets in triplicates or quadruplicates were pre-incubated for 60 min in RPMI-1640 containing 3.3 mmol/l glucose, then consecutively incubated at 3.3 mmol/l and 16.7 mmol/l glucose for 1 hr at 37°C in 200 µl modified Krebs-Ringer bicarbonate buffer containing 20 mmol/l HEPES and 0.25% BSA (KRBH-BSA). Medium was collected, centrifuged, and frozen at −20°C and islets were lysed using 0.1% BSA-GB/NP-40. Insulin in medium and islet lysates was determined by ELISA.

## Immunofluorescence staining and analysis

Pancreases were fixed with zinc-formalin (neonates) or 4% buffered formaldehyde (weaning and adults) for 3 hr. Paraffin sections (5 µm thick) were rehydrated and antigen retrieval was performed using a Biocare pressure cooker and citrate buffer (pH = 6). The following antibodies were used: guinea pig anti-insulin 1:200 (DakoCytomation, Glostrup, Denmark), rabbit anti-Ki67 1:200 (Thermo Scientific, Kalamazoo, MI), goat anti PDX-1 1:200 (kindly provided by Dr. C. V. Wright, Vanderbilt University, TN), mouse anti-NKX6.1 1:200 (Developmental Studies Hybridoma Bank), PS6 (Cell Signaling, MA), mouse anti E-cadherin 1:100 (BD Biosciences, NJ), mouse anti PCNA 1:500 (DakoCytomation, Glostrup, Denmark) rabbit anti H3P 1:100 (Cell Signaling, MA). TUNEL staining was performed with the Roche Cell Death Detection Kit (Roche Diagnostics), cell nuclei were visualized with DAPI staining. Secondary antibodies are all from Jackson Immuno Research Laboratories. Digital images of pancreatic islets were obtained with a Zeiss LSM-710 and Nikon A1R confocal microscope using a x40 oil objective. For analysis of β-cell proliferation and apoptosis, β-cells were counted using Adobe Photoshop CS6 software.

To determine β-cell mass, consecutive paraffin sections 75 µm (in young and adult mice) or 50 µm (in newborns) apart spanning the entire pancreas were stained for insulin and hematoxylin. Digital images were obtained at an original magnification of × 4 with a Nikon C1 confocal microscope, stitched using NIS-Elements software (Nikon, Melville, NY), and the percent area covered by insulin was determined. β-Cell mass was calculated as the product of pancreas weight and percentage insulin area.

## Western blot

Protein levels were assessed using antibodies against: total and phospho S6 ribosomal protein (Ser240/244), insulin receptor substrate 2 (IRS2), total and phospho-AKT/protein kinase B (Ser473 and Thr 308), phospho-4EBP-1(Thr37/46), BiP, PDX-1, tubulin, and Hsp90. Peroxidase-conjugated AffiniPure goat anti-rabbit, anti-chicken and anti-mouse IgG from Jackson ImmunoResearch Laboratories (West Grove, PA) were used as secondary antibodies.

## Quantitative real-time RT-PCR

RNA was extracted using TRI Reagent (Biolab, Jerusalem, Israel) and an RNeasy Micro Kit (Qiagen); samples of 260 ng total RNA were reverse transcribed using a high capacity cDNA Reverse Transcription Kit (qScript, Quantabio, Beverly, MA). Quantitative real-time RT-PCR for total and spliced Xbp1 was performed on a Prism 7000 Sequence Detection System using the Power SYBR Green PCR Master Mix (Applied Biosystems, Foster City, CA). All samples were corrected for glyceraldehyde-3-phosphate dehydrogenase. The following oligonucleotides were used for the PCR of total and spliced Xbp1: forward T-Xbp1, 5'- AAGAACACGCTTGGGAAT-3' and reverse t-Xbp1: 5'- ACTCCCCTTGGCCTCCAC-3; forward s-Xbp1: 5'-GAGTCCGCAGCAGGTG-3' and reverse s-Xbp1: 5'-GTGTCAGAGTCCATGGGA-3'.

## RNAseq

RNA sequencing libraries were constructed from 120 ng of total RNA using the TruSeq RNA V2 sample prep kit (Illumina). Single read sequencing was performed on Illumina hiSeq2500 to 50 bp. Reads were aligned to the mouse genome GRCm38 using STAR (v2.5.2b). Quantification of read counts

per gene was performed using htseq-count (version 0.7.2) and differentially expressed genes were identified using DESeq2 package (version 1.12.4) for normalization and evaluation of differential expression. The significance threshold for comparisons was taken as p value < 0.05. Gene set enrichment analyses were done using Genomica (http://www.genomespace.org) and GSEA (http://software.broadinstitute.org/gsea/index.jsp) and pathway analyses were carried out using the software Ingenuity Pathway Analysis (IPA; Ingenuity Systems, http://www.ingenuity.com).

## Statistical analysis

Statistical analysis was performed using GraphPad Prism 6.01 software (GraphPad Software, La Jolla, CA). Differences between multiple groups were analyzed by one-way ANOVA. Two-tailed paired Student's t test was used to compare differences between two groups. One-sample Student's t test was performed to validate statistical differences in experiments expressing data as relative to control. Data in graphs and tables are presented as means ±SEM (standard error of the mean). $p < 0.05$ was considered significant.

# Additional information

### Funding

| Funder | Grant reference number | Author |
|---|---|---|
| Israel Science Foundation | ISF-347/12 | Gil Leibowitz |
| Israel Science Foundation | ISF-1563/14 | Gil Leibowitz |
| Israel Science Foundation | 2323/17 | Gil Leibowitz |

The funders had no role in study design, data collection and interpretation, or the decision to submit the work for publication.

### Author contributions

Yael Riahi, Conceptualization, Formal analysis, Validation, Methodology, Writing—review and editing; Tal Israeli, Formal analysis, Methodology, Writing—review and editing; Roni Yeroslaviz, Shoshana Chimenez, Marina Sebag, Nava Polin, Formal analysis; Dana Avrahami, Formal analysis, Methodology; Miri Stolovich-Rain, Ernesto Bernal-Mizrachi, Yuval Dor, Conceptualization, Writing—review and editing; Ido Alter, Formal analysis, Writing—original draft, Writing—review and editing; Erol Cerasi, Conceptualization, Data curation, Writing—original draft, Writing—review and editing; Gil Leibowitz, Conceptualization, Supervision, Funding acquisition, Writing—original draft, Writing—review and editing

### Author ORCIDs

Tal Israeli (iD) http://orcid.org/0000-0001-9293-0827
Erol Cerasi (iD) http://orcid.org/0000-0002-8234-3618
Gil Leibowitz (iD) http://orcid.org/0000-0002-6915-4361

### Ethics

Animal experimentation: This study was performed in strict accordance with the recommendations in the Guide for the Care and Use of Laboratory Animals of the Hebrew University. All of the animals were handled according to approved institutional animal care and use committee of the Hebrew University. The protocol was approved by the Committee on the Ethics of Animal Experiments of the Hebrew University (Permit Number: MD-17-15065-4). Every effort was made to minimize animal suffering.

### Decision letter and Author response

Decision letter https://doi.org/10.7554/eLife.38472.032
Author response https://doi.org/10.7554/eLife.38472.033

## Additional files

### Supplementary files
• Transparent reporting form
DOI: https://doi.org/10.7554/eLife.38472.022

### Data availability
The RNA-seq data is available through NCBI. The accession number is: GSE114927

The following dataset was generated:

| Author(s) | Year | Dataset title | Dataset URL | Database and Identifier |
|---|---|---|---|---|
| Riahi Y, Israeli T, Yeroslaviz R, Chimenez S, Avrahami D, Stolovich-Rain M, Alter I, Sebag M, Polin N, Bernal-Mizrachi E, Dor Y, Cerasi E, Leibowitz G | 2018 | RNAseq analysis of whole islets from pre-weaning wild type and Akita mice | https://www.ncbi.nlm.nih.gov/geo/query/acc.cgi?acc=GSE114927 | Gene Expression Omnibus, GSE114927 |

The following previously published datasets were used:

| Author(s) | Year | Dataset title | Dataset URL | Database and Identifier |
|---|---|---|---|---|
| Helman A, Klochendler A, Azazmeh N, Gabai Y, Horwitz E, Anzi S, Swisa A, Condiotti R, Granit RZ, Nevo Y, Fixler Y, Shreibman D, Zamir A, Tornovsky-Babeay S, Dai C, Glaser B, Powers AC, Shapiro AM, Magnuson MA, Dor Y, Ben-Porath I | 2016 | RNA profiling of P16ink4a-expressing pancreatic beta-cells | https://www.ncbi.nlm.nih.gov/geo/query/acc.cgi?acc=GSE76992 | Gene Expression Omnibus, GSE76992 |
| Taylor BL, Liu FF, Sander M | 2013 | Identification of Nkx6.1 regulated genes in mature pancreatic islets | https://www.ncbi.nlm.nih.gov/geo/query/acc.cgi?acc=GSE40470 | Gene Expression Omnibus, GSE40 470 |
| Sachdeva MM, Claiborn KC, Khoo C, Yang J, Groff DN, Mirmira RG, Stoffers DA | 2009 | Chromatin immunoprecipitation of mouse MIN6 pancreatic beta cells to identify Pdx1 targets | https://www.ebi.ac.uk/arrayexpress/experiments/E-MTAB-134/ | ArrayExpress Archive of Functional Genomics Data, E-MTAB-134 |

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
