## [Decision Letter]

Thank you for submitting your article "Inhibition of mTORC1 by ER stress impairs neonatal β-cell expansion and predisposes to diabetes" for consideration by *eLife*. Your article has been reviewed by two peer reviewers, and the evaluation has been overseen by a Reviewing Editor and Harry Dietz as the Senior Editor. The reviewers have opted to remain anonymous.

The reviewers have discussed the reviews with one another and the Reviewing Editor has drafted this decision to help you prepare a revised submission.

This manuscript from Riahi and co-authors describes an elegant study, which has unfortunately been hastily written up, that examines the mechanism of the β-cell mass deficit in male mice carrying the *Ins2*^C96Y^ mutation on a B6 background (referred to in this paper as simply *Akita* mice) in a detailed and careful manner and suggest that mTORC1 may modify the phenotype. The authors clearly show an important role for a profound β-cell proliferation defect in young *Akita* mice. They also evaluate cell size and β-cell function. The authors identify a loss of mTORC1 signaling as a critical mechanism blocking proliferation. Using a negative regulator of mTORC1, *Tsc1*, they are able to dissect the components of the phenotype that are mTORC1-dependent.

This is an important study that adds to a growing body of literature defining pathogenically relevant links between β-cell stress and failure of adaptive expansion and maturation.

Essential revisions:

1) The reviewers and editors have concerns regarding the small group sizes which have been used for many studies (n=3) and the potential effect of "outliers". The authors should increase the number of mice studied and ensure that all data points are shown.

2) The authors also fast mice for 16h before glucose tolerance testing which leads to a catabolic state and weight loss (Ayala et al., 2010). 4-6 hours is much better. The authors must justify the decision to fast the mice for this prolonged period.

3) These authors suggest that the effect of insulin mutations in this mouse model are on postnatal development/function of β-cells whereas the accompanying paper by Balboa et al. in a human model suggest probably prenatal. It is not clear if this reflects a difference between mice and men or something else.

4) The current work leaves open some key questions, including the mechanisms linking ER stress to changes in maturation. Exactly how do the authors propose mTORC1 is controlled by ER-stress in their model? This should be discussed in more detail, perhaps drawing on the signalling networks that can be constructed from open data.

---

## [Author Response]

Essential revisions:1) The reviewers and editors have concerns regarding the small group sizes which have been used for many studies (n=3) and the potential effect of "outliers". The authors should increase the number of mice studied and ensure that all data points are shown.

We performed additional experiments and increased the n as detailed below: in Figure 1—figure supplement 3A, we added blood glucose values that were recorded in WT and *Akita* newborns: the number of samples was increased from 3 to 8 in the wildtype group and to 7 in the *Akita* group. In Figure 1—figure supplement 3B, we analyzed β-cell mass in 2 additional newborn (P1-2) *Akita* and wildtype mice; we now have n=5 in each group (Figure 3E and Figure 1—figure supplement 3B were updated). The numbers of animals and cells used for quantifications were added in the legend to all figures. In Figure 5, we increased the number of mice used for the quantification of phosphorylated S6 (n=4-6 mice in each group). In addition, in Figure 5—figure supplement 2, we performed additional experiments and increased the number of TUDCA treated and control mice; the revised manuscript shows the results of n=4-6 mice in each group. In Figure 3, the number of experiments indicated in the legend remained from a previous version of the manuscript and was incorrect. The correct number (n=6) is shown in the revised manuscript. In Western blotting and RNA-seq experiments, analyses were performed on pools of 400-600 islets from many wildtype and *Akita* mice in each experiment, therefore the results represent the mean of multiple animals of each strain.

2) The authors also fast mice for 16h before glucose tolerance testing which leads to a catabolic state and weight loss (Ayala et al., 2010). 4-6 hours is much better. The authors must justify the decision to fast the mice for this prolonged period.

In light of this comment, we performed an experiment in which IPGTT was performed in the same animals after 4 and 16 h fast in wildtype and in *Akita* mice. The findings are shown in Author response image 1. Note that at 16h, fasting blood glucose of wildtype mice is lower than that at 4h (81 compared to 156.5 mg/dl), indicating that at 4h glucose is not at steady-state. Moreover, at 4h glucose excursion following IP glucose injection is small, therefore it is difficult to assess glucose disposal. In addition, in *Akita* mice, fasting blood glucose at 4h is much higher than in wildtype mice, therefore it is difficult to compare the glucose response between diabetic and non-diabetic animals. Moreover, in diabetic animals blood glucose levels were >600 mg/dl (above the detection limit of the glucometer), thus the animals are exposed to extreme hyperglycemia. Overall, we believe that these findings suggest that in our model, overnight fast is superior to 4h for the assessment of the glucose response.

**Author response image 1. respfig1:** Effects of 4 and 16h fast on IPGTT. Wildtype (WT) and *Akita* mice were fasted for 4 or 16h followed by IPGTT (1.5 gr/kg glucose) (n=4 in each group).

3) These authors suggest that the effect of insulin mutations in this mouse model are on postnatal development/function of β-cells whereas the accompanying paper by Balboa et al. in a human model suggest probably prenatal. It is not clear if this reflects a difference between mice and men or something else.

We believe that the differences are explained by using different experimental systems. In vitro differentiation of iPS cells is a complex process, which may not completely overlap pre- and postnatal differentiation in vivo. The stages of iPS differentiation to β-cells have similarities with β-cell differentiation in vivo; however there might be cellular heterogeneity in the differentiation state at a given time point. Humans with neonatal diabetes due to insulin mutations are born with normal body weight and are normoglycemic at birth, indicating that hypoinsulinemia and most probably the decrease in β-cell mass develops during the postnatal period, therefore it is unlikely that the apparent dissimilarities reflect a difference between mice and men.

4) The current work leaves open some key questions, including the mechanisms linking ER stress to changes in maturation. Exactly how do the authors propose mTORC1 is controlled by ER-stress in their model? This should be discussed in more detail, perhaps drawing on the signalling networks that can be constructed from open data.

Our data indicate that ER stress inhibits mTORC1 through inhibition of insulin/IGF-1 signaling at the level of PI3 kinase-Akt (see Figure 5—figure supplement 1). We found that ER stress induced by the *Akita* proinsulin mutant or by treatment with thapsigargin had modest or no effect on IRS2 expression, while markedly inhibiting Akt phosphorylation at Serine 473 and Threonine 308 that are both required for its activity. Our preliminary data show that the expression of P85α, an important component of the PI3 kinase which is required to its activity, is decreased. We are currently performing in depth study on how different types of ER stress affect P85α mRNA and protein level and subcellular localization. These extensive studies are ongoing; at this stage they are too premature to be included in the manuscript. We extended the discussion on possible mechanisms leading to inhibition of Akt-mTORC1 signaling based on the literature.

References

Ayala, J.E., Samuel, V.T., Morton, G.J., Obici, S., Croniger, C.M., Shulman, G.I., Wasserman, D.H., McGuinness, O.P. (2010). Standard operating procedures for describing and performing metabolic tests of glucose homeostasis in mice. Disease Models & Mechanisms, 3: 525-534; doi: 10.1242/dmm.006239.